# Getting More Juice Out of the SFT Data: Reward Learning from Human Demonstration Improves SFT for LLM Alignment

**Jiaxiang Li**
University of Minnesota
Minneapolis, MN, USA
`li003755@umn.edu`

**Siliang Zeng**
University of Minnesota
Minneapolis, MN, USA
`zeng0176@umn.edu`

**Hoi-To Wai**
Chinese University of Hong Kong
Hong Kong
`htwai@se.cuhk.edu.hk`

**Chenliang Li**
Texas A&M University
College Station, TX, USA
chenliangli@tamu.edu

**Alfredo Garcia**
Texas A&M University
College Station, TX, USA
alfredo.garcia@tamu.edu

**Mingyi Hong**
University of Minnesota
Minneapolis, MN, USA
`mhong@umn.edu`

## Abstract

Aligning human preference and value is an important requirement for contemporary foundation models. State-of-the-art techniques such as Reinforcement Learning from Human Feedback (RLHF) often consist of two stages: 1) supervised fine-tuning (SFT), where the model is fine-tuned by learning from human demonstration data; 2) Preference learning, where preference data is used to learn a reward model, which is in turn used by a reinforcement learning (RL) step to fine-tune the model. Such reward model serves as a proxy to human preference, and it is critical to guide the RL step towards improving the model quality. In this work, we argue that the SFT stage significantly benefits from learning a reward model as well. Instead of using the human demonstration data directly via supervised learning, we propose to leverage an Inverse Reinforcement Learning (IRL) technique to *simultaneously* build an reward model and a policy model. This approach leads to new SFT algorithms that are not only efficient to implement, but are robust to the presence of low-quality supervised learning data. Moreover, we discover a connection between the proposed IRL based approach, and a recent line of works called Self-Play Fine-tune (SPIN, Chen et al. [2024]). Theoretically, we show that the proposed algorithms converge to the stationary solutions of the IRL problem. Empirically, we align 1B and 7B models using proposed methods and evaluate them on a reward benchmark model and the HuggingFace Open LLM Leaderboard. The proposed methods show significant performance improvement over existing SFT approaches. Our results indicate that it is beneficial to leverage reward learning throughout the entire alignment process. Our code is available at https://github.com/JasonJiaxiangLi/Reward_learning_SFT.

## 1 Introduction

Large Language Models (LLMs) have become the cornerstone of modern artificial intelligence applications. They are believed to lead the way towards artificial general intelligence [Bubeck et al., 2023], also have shown great capabilities towards specialized domains such as math problem solving [Cobbe et al., 2021, Trinh et al., 2024, Wei et al., 2022, Lewkowycz et al., 2022], code generation [Chen et al., 2021, Austin et al., 2021, Li et al., 2022], text generation [Anil et al., 2023, Touvron et al., 2023, Thoppilan et al., 2022], etc. Usually, researchers need to align the pre-trained LLMs

38th Conference on Neural Information Processing Systems (NeurIPS 2024).

with certain exquisitely prepared human-labeled data to achieve desired performance over certain tasks, a process which is thus known as alignment or fine-tuning. The alignment datasets can be categorized into two classes: the demonstration data, with the input prompt and a human response; and the preference data, with the input prompt and two responses, where human labeler will pick a chosen one and a rejected one. With the alignment datasets, one could employ methods like supervised fine-tune (SFT, Ouyang et al. [2022], Tunstall et al. [2023], Chung et al. [2024]) for aligning demonstration datasets, and reinforcement learning from human feedback (RLHF, Christiano et al. [2017], Ouyang et al. [2022]) and direct preference optimization (DPO, Rafailov et al. [2024]) for aligning preference datasets. More specifically, RLHF *explicitly* trains a reward model and uses reinforcement learning (in particular, policy optimization) methods to obtain a fine-tuned version of the LLM; on the other hand, DPO and many of its extensions simplifies the RLHF by training the LLM policy model directly, while *implicitly* learns the reward model via log of the ratio of likelihood between the learned model and a reference model. In practice, both types of methods exhibit better performance over SFT on the demonstration datasets, and they are adopted by state-of-the-art LLMs, for example ChatGPT benefited from RLHF (see Ouyang et al. [2022]), zephyr benefited from DPO (see Tunstall et al. [2023]).

It is interesting to observe that, when dealing with preference data, state-of-the-art methods usually build an (explicit or implicit) reward model to evaluate the quality of responses for a given prompt. On the contrary, typically no reward modeling is done for demonstration datasets. Why this is the case? One may argue that, for a given set of prompts, preference datasets contain explicit preference information which is not found in the demonstration datasets; since this kind of information is harder to extract, it motivates the use of complicated methods such as reward modeling. However, since *human preferences* are also implicit in the demonstration data, one can argue that training a reward model that encodes human value distilled from these datasets may help to boost the alignment capability of the LLM. Indeed, in the RL literature, it is known that if the agents are given a set of demonstration data, then the so-called inverse RL methods (which learns the reward and policy simultaneously) can outperform the behavior cloning methods (which corresponds to supervised fine-tune in LLM alignments) by a large margin. In a Markov decision process (MDP), it is likely that supervised learning methods which naively fit the demonstration data will suffer from the distribution shift problem – the fine-tuned policy from supervised learning can produce unsatisfactory generations in certain states which were unseen in the training dataset [Ross et al., 2011]. Through formulating the learning from demonstration problem in a MDP setting, typical inverse reinforcement learning methods [Ziebart et al., 2008, Ross et al., 2011, Zeng et al., 2022] can alleviate such distribution shift issues. Witnessing the success in ChatGPT, where the alignment of LLMs is modelled in the MDP setting due to the auto-regressive process, one would expect that the LLM alignment with demonstration datasets can be improved as well through deploying imitation learning / inverse reinforcement learning methods.

Inspired by the above observation, we pose the following question:

> Does building a reward model using the demonstration data benefit the alignment process?

**Contribution of this work.** This paper answers the above question affirmatively. Specifically, by developing a framework based on certain IRL technique, we show that building a reward from demonstration datasets can significantly improve the quality of the resulting model, as compared to that obtained by standard reward-free SFT (1). Our main contributions are listed as below:

- We develop a new reward-based SFT approach, which takes the form of a *bilevel* optimization, where in the *lower-level*, LLM policy is learned via policy optimization for a given reward, while in the *upper-level*, the reward model is optimized so to maximize the likelihood for observing the demonstration data.

- Based on the above formulation, we propose two alignment algorithms, one learns the reward model explicitly, and the other implicitly. For the first algorithm, we show that the reward learned from only demonstration data already possesses strong capabilities in distinguishing between chosen and rejected responses; see Figure 1 and our experiment for details. For the second algorithm, we made an interesting observation that implicitly learning a reward is equivalent to improving the model by comparing the demonstration data with the *synthetic* data generated by the past models. Somewhat surprisingly, the resulting algorithm is closely related to the self-play fine-tune (SPIN, Chen et al. [2024]) algorithm, recently proposed from a completely different viewpoint. It is worth pointing out that unlike SPIN, our proposed algorithms have finite-time convergence guarantees.

- We demonstrate the power of the proposed approach theoretically and numerically. We prove that our implicit reward learning algorithm converges to some stationary point of our proposed formulation. We show that the proposed algorithms outperform vanilla SFT in almost all cases we have tested, for example the model performance on HuggingFace Open LLM Leaderboard increases from 59.47% to 61.03%. To our knowledge, this is the first work that formally demonstrate the power of reward learning when dealing with demonstration data for LLM alignment.

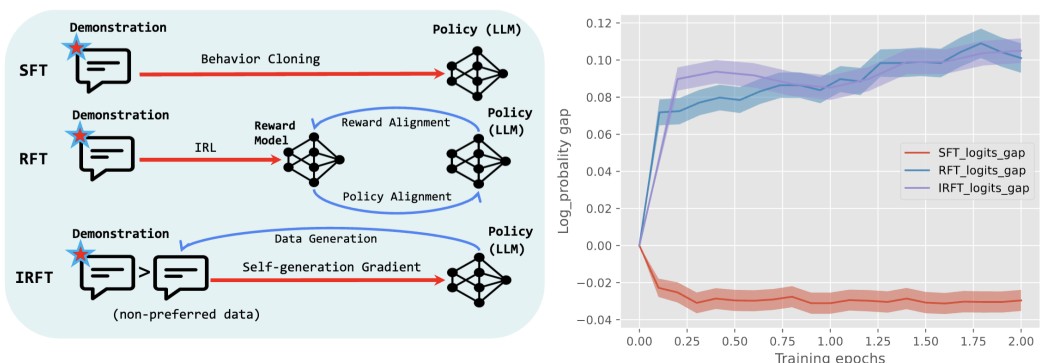

Figure 1: **Left:** Difference between SFT and the two proposed methods: RFT (Algorithm 1) and IRFT (Algorithm 2); **Right:** Log probability gap between the chosen/preferred continuation and the rejected/non-preferred continuations for different methods. All methods *only* consume the chosen/preferred data, but RFT and IRFT can effectively distinguish between chosen and rejected continuations; see Example 2 in Sec. 3 for the detailed settings.

**Notations.** We use $\pi(y|x)$ to denote the LLM output probability for continuation $y$ with input prompt $x$, and we refer to $\pi$ as the policy. We use the notation $\pi(y|x; \boldsymbol{\theta})$ if the model $\pi$ is directly parameterized by parameters $\boldsymbol{\theta}$. For the case when $\pi$ is indirectly determined by parameter $\boldsymbol{\theta}$, we use notation $\pi_{\boldsymbol{\theta}}(y|x)$. We use $\mathcal{D} = \{(x, y)\}$ to denote the demonstration dataset and $\mathcal{P} = \{(x, y_w, y_l)\}$ for the preference dataset, where $y_w$ is preferred over $y_l$. Since we assume that the demonstration continuations $y$ are collected from a human expert distribution, we also denote $(x, y) \sim \mathcal{D}$ as $x \sim \rho, y \sim \pi^E(\cdot|x)$ when taking the expectations, where $\rho$ is the distribution of the input prompts when collecting the data. We similarly have the notation $x \sim \rho, (y_l \prec y_w) \sim \pi^P(\cdot|x)$ for the preference dataset.

## 2 Preliminaries

Consider a Large Language Model (LLM) parameterized by $\boldsymbol{\theta}$ and denote the output probability by $\pi(y|x; \boldsymbol{\theta})$ where $x = [x_1, ..., x_n]$ is the sequence of input prompts and $y = [y_1, ..., y_m]$ is the sequence of output continuation. Typical LLM is an auto-regressive model, meaning that it predicts the output probability of the $y_j$ given all tokens in $x$ and $y_{<j} := [y_1, ..., y_{j-1}]$ ($y_{<1}$ is null), i.e.

$$\pi(y|x; \boldsymbol{\theta}) = \prod_{j=1}^{m} \pi(y_j|x, y_{<j}; \boldsymbol{\theta}).$$

In this paper, we do not focus on the architecture design of LLMs. We will fix the LLM architecture and always denote it as a probability model $\pi(y|x; \boldsymbol{\theta})$. The following discussions review two common procedures for fine-tuning $\boldsymbol{\theta}$: (1) supervised fine tuning (SFT) over demonstration dataset, (2) reinforcement learning with human feedback (RLHF) over preference dataset that consists of two steps: LLM alignment/fine-tuning based on a reward model using policy optimization; and reward learning process to learn the optimal reward for the preference dataset.

**SFT.** Given a *demonstration dataset* $\mathcal{D} := \{(x, y)\}$, the SFT optimizes the following problem:

$$\max_{\boldsymbol{\theta}} \ \ell_{\text{SFT}}(\boldsymbol{\theta}) := \mathbb{E}_{(x,y) \sim \mathcal{D}} \left[ \log \pi(y|x; \boldsymbol{\theta}) \right]. \tag{1}$$

It is easy to see that the above problem shares the same optimal solutions as $\min_{\boldsymbol{\theta}} \ \mathbb{E}_{x \sim \rho}[D_{\text{KL}}(\pi^E(\cdot|x) \| \pi(\cdot|x; \boldsymbol{\theta}))]$. The latter shows that SFT aims at imitating the demonstration dataset via minimizing the KL divergence. It is worth noting that the SFT stage described

here is closely related to the imitation learning approach used in the RL literature for learning from demonstration [Osa et al., 2018], whose goal is to mimic the policy of an expert.

**RLHF.** Suppose that we have a reward model $r(x, y; \phi)$ (parameterized by $\phi$ and to be defined later) for any given input and output pair $(x, y)$, the LLM can be fine tuned by the following RL problem:

$$\max_{\boldsymbol{\theta}} \ell_{\mathrm{RL}}(\boldsymbol{\theta}) := \mathbb{E}_{x \sim \rho, y \sim \pi(\cdot|x;\boldsymbol{\theta})} \left[ r(x, y; \phi) \right] - \mathbb{E}_{x \sim \rho} [D_{\mathrm{KL}}(\pi(\cdot|x; \boldsymbol{\theta}) \| \pi_{\mathrm{ref}}(\cdot|x))], \quad (2)$$

where $\pi_{\mathrm{ref}}$ is a fixed reference model. Note that the KL regularization term in (2) is not computable given the sheer amount of possible output $y$ (which could be *corpus_size$^{max\_sequence\_length}$* in most language model tasks), therefore (2) is usually solved by standard policy optimization techniques such as REINFORCE [Ahmadian et al., 2024] or PPO [Schulman et al., 2017].

To find an appropriate reward model $r(x, y; \phi)$, RLHF (see e.g., Christiano et al. [2017]) leverages a set of *preference dataset* $\mathcal{P} := \{(x, y_w, y_l)\}$, where each data contains a pair of output $y_w, y_l$, and $y_w$ is preferred over $y_l$ by human labeler (denoted as $y_w \succ y_l$). The Bradley-Terry model [Bradley and Terry, 1952] assumes that the probability of choosing $y_w$ over $y_l$ is

$$\mathbb{P}(y_w \succ y_l \mid x) = \frac{\exp(r(y_w; x))}{\exp(r(y_w; x)) + \exp(r(y_l; x))} = \sigma\left(r(y_w; x) - r(y_l; x)\right).$$

One could formulate the following problem to find the reward model:

$$\max_{\boldsymbol{\phi}} \ell_{\mathrm{RM}}(\boldsymbol{\phi}) := \mathbb{E}_{x \sim \rho, (y_l \prec y_w) \sim \pi^P(\cdot|x)} \left[ \log\left( \sigma\left(r(x, y_w; \boldsymbol{\phi}) - r(x, y_l; \boldsymbol{\phi})\right)\right) \right]. \quad (3)$$

It is widely observed in the literature that, models trained via episodically learning the policy (2) and learning the reward (3) typically outperforms those that are only trained using SFT [Ouyang et al., 2022]. The reward model guides the performance of the LLM and allows a better generalization ability via the consistent input of the preference data from human labeler. Follow up works such as DPO proposes to incorporate reward learning implicitly by utilizing the structure of the optimal solution of the RL problem (2); for more details about the DPO, see Rafailov et al. [2024].

**Discussion.** At this point, let us take a step back and think about the above process. The LLM alignment problem takes human labeled demonstration and preference data to produce an *aligned* model. Clearly, both kinds of data encode information about how human would like the LLM output to be, but the processes of extracting such information is quite different (i.e., supervised learning vs RL). A series of questions naturally arises: Is supervised learning the best way to extract human inclination from the demonstration data? Can we also learn a reward model from the demonstration data to gauge human preference? Will policy model learned via such reward improve the supervised learning approach? In the next section, we will dive deep to carefully address these questions.

# 3 Reward Learning and Policy Fine Tuning from Demonstration Data

In this section, we argue that reward learning from the demonstration dataset can benefit the LLM alignment problem. To do so, we develop a joint reward learning and policy fine tuning formulation and understand its capabilities in improving the LLM policy. The new formulation inspired us to develop two reward learning paradigms: *i)* Explicit reward learning, where a (parameterized) reward model is learned together with the language model policy, and *ii)* Implicit reward learning, where the reward model is learned implicitly through directly optimizing the policy, avoiding learning *two* models simultaneously.

## 3.1 Joint Reward-learning and Policy Fine-tuning by Inverse RL

A challenge with learning only from the demonstration dataset is that the Bradley-Terry model (3) can no longer be used due to the lack of *pairs* of preference data. However, all is not lost as we recall that it is the *value* of the reward model that should be used in the fine-tuning process (2). Therefore, with only demonstration data $\mathcal{D}$, a reasonable formulation is to combine the supervised learning problem (1) with the optimal policy generation problem (2), by requiring that the generated policy to 'match' with the demonstration dataset. With this intuition in mind, we consider the *joint* reward and policy learning problem via a maximum likelihood inverse reinforcement learning (ML-IRL) formulation [Ziebart et al., 2008, 2013, Zeng et al., 2022]:

$$\max_{\boldsymbol{\theta}} \ell(\boldsymbol{\theta}) := \mathbb{E}_{x \sim \rho, y \sim \pi^{\mathrm{E}}(\cdot|x)} \left[ \log \pi_{\boldsymbol{\theta}}(y \mid x) \right]$$

$$\text{s.t. } \pi_{\boldsymbol{\theta}} := \arg\max_{\pi} \mathbb{E}_{x \sim \rho, y \sim \pi(\cdot|x)} \left[ r(x, y; \boldsymbol{\theta}) - \beta D_{\mathrm{KL}}\left(\pi(\cdot|x) \| \pi_{\mathrm{ref}}(\cdot|x)\right) \right]. \quad (4)$$

The above problem has a *bilevel* structure which trains a reward model $r(x, y; \boldsymbol{\theta})$. At the upper level, its objective is similar to that of SFT (1), but is evaluated on the policy $\pi_{\boldsymbol{\theta}}$ induced by the reward model $r(x, y; \boldsymbol{\theta})$; meanwhile, this policy $\pi_{\boldsymbol{\theta}}$ is found in the lower level using the RL objective (2).

There are several advantages of the bilevel formulation (4) over standard SFT (1). First, we notice formulating SFT as a RL / IRL problem can alleviate distribution shift and improve the generalization power [Ross et al., 2011]. In fact, we observe that (4) tends to give a less extreme policy even when the demonstration dataset is extreme. The latter is observed in the following stylized example.

**Example 1.** Suppose we have only one state (input prompt) $x$ and three actions (continuations) $y_1, y_2, y_3$. Let the reference model $\pi_{\text{ref}}$ be a uniform distribution over all continuations, and the demonstration dataset is $\mathcal{D} = \{y_3\}$. One could easily compute the optimal solution for (1) and (4) by first-order optimality conditions. From Table 1 we can see that SFT (imitation learning) pushes all the likelihood toward the demonstration dataset, whereas ML-IRL (4) maintains non-zero weights for unseen data in the demonstration datasets. This is particular useful when we want to fine-tune from a pre-trained model, which is presumed to be powerful and have useful information already. □

Second, since the lower level problem in (4) encapsulates a generation process, it is anticipated that the proposed method can better distinguish between the preferred and non-preferred data than SFT, even if it is only trained on the demonstration dataset. The following numerical example highlights this point:

**Example 2.** We compare the solution of SFT (1) and IRL (4) numerically, where the latter is solved using two algorithms RFT and IRFT (to be introduced shortly). We choose the preference based dataset Anthropic-HH and only keep the preferred continuation to form a demonstration dataset $\tilde{\mathcal{D}} = \{(x, y_w)\}$ to implement SFT and IRL. We then compute the log probability

| Action | $y_1$ | $y_2$ | $y_3$ |
|---|---|---|---|
| $\pi_{\text{ref}}$ | 0.33 | 0.33 | 0.33 |
| $\mathcal{D}$ | | $\{y_3\}$ | |
| $\pi_{\text{SFT}}$ | 0.0 | 0.0 | 1.0 |
| $\pi_{\text{IRL}}$ | $\frac{2}{2+e^{R/\beta}}$ | $\frac{2}{2+e^{R/\beta}}$ | $\frac{e^{R/\beta}}{2+e^{R/\beta}}$ |

Table 1: A state-less counter-example with three actions where IRL-based fine-tune (4) shows regularization effect over SFT (1) to maintain weights over unseen data in the demonstration dataset $\mathcal{D}$. Here we assume $r \in [0, R]$.

gap $\log(\pi(y_w|x)) - \log(\pi(y_l|x))$ between the preferred $y_w$ and non-preferred $y_l$ on the test dataset; see Figure 1 right side. We observe that although all three methods are not exposed to the non-preferred data $y_l$ during the training process, the IRL-based methods effectively distinguish the preferred continuation over the non-preferred one, while SFT assigns larger probability to the non-preferred continuation (see Section 5 for the details of the implementation).

Comparing to SFT (1), the bilevel problem (4) appears to be more complicated. In particular, solving standard bilevel optimization problem typically involves computation of Hessian matrices, which is too expensive for LLM related applications [Liu et al., 2023]. Fortunately, in our next result, we show that the bilevel problem can be significantly simplified (proof in Appendix B):

**Lemma 3.1.** *Problem* (4) *is equivalent to the following minimax optimization problem:*

$$\max_{\boldsymbol{\theta}} \min_{\pi} \mathbb{E}_{x \sim \rho, y \sim \pi^{\text{E}}(\cdot|x), \tilde{y} \sim \pi(\cdot|x)} \left[ \frac{r(x, y; \boldsymbol{\theta}) - r(x, \tilde{y}; \boldsymbol{\theta})}{\beta} + D_{\text{KL}}\Big(\pi(\cdot|x) \| \pi_{\text{ref}}(\cdot|x)\Big) \right]. \quad (5)$$

The above reformulation is remarkable. First, minimax problem is much easier to solve as compared with bilevel problem, e.g., a simple alternating minimization can yield reasonably good solution; see Algorithm 1 for such an algorithm, and Sec. 3.3 for its theoretical analysis. **More importantly**, it shows that even only the demonstration data is available, the reward optimization problem takes a similar form as what has been used in RLHF (3), where not one but *two* reward functions are contrasted. The key difference here is that one reward is evaluated on the continuation $y$ in $\mathcal{D}$, the other is evaluated on $\tilde{y}$, which is the continuation *generated* from the current policy $\pi(\cdot|x)$. We believe that such contrast is the key reason that enables the IRL based formulation to distinguish the preferred continuations over the non-preferred ones; see Example 2 and Figure 1.

Now that we have turned the original bilevel problem (4) into a minimax optimization problem (5), we can naturally develop a gradient-descent-ascent type algorithm for (5), which alternates between updating the policy according to the current reward, and updating the reward based on the current policy — an algorithm that we call *Reward-learning Fine-tune* (RFT), see Algorithm 1. Note that in

---

**Algorithm 1:** *Reward-learning Fine-Tune* (RFT)

---

**Input:** Initialize reward parameter $\boldsymbol{\theta}_0(\boldsymbol{\theta}_{-1,K} = \boldsymbol{\theta}_0)$ and policy model $\pi^0$, the stepsize of reward update $\eta_t$, and $T, K$ the outer and inner iterations.

**for** $t = 0, 1, \ldots, T-1$ **do**

    Take $\boldsymbol{\theta}_{t,0} = \boldsymbol{\theta}_{t-1,K}$

    **Data Sample:** Sample state $x_{t,k} \sim \rho$, an expert response $y_{t,k} \sim \pi^{\mathrm{E}}(\cdot|x_{t,k})$ and agent response $\tilde{y}_{t,k} \sim \pi^t(\cdot|x_{t,k})$, for $k = 0, 1, ..., K-1$

    **for** $k = 0, 1, ..., K-1$ **do**

        **Estimate Gradient:** Calculate the stochastic gradient $g_{t,k}$ w.r.t. $\boldsymbol{\theta}$ via

        $g_{t,k} = \frac{1}{\beta}\nabla_{\boldsymbol{\theta}} r(x_{t,k}, y_{t,k}; \boldsymbol{\theta}_{t,k}) - \frac{1}{\beta}\nabla_{\boldsymbol{\theta}} r(x_{t,k}, \tilde{y}_{t,k}; \boldsymbol{\theta}_{t,k})$

        **Reward Alignment:** $\boldsymbol{\theta}_{t,k+1} := \boldsymbol{\theta}_{t,k} + \eta_t g_{t,k}$

    **end for**

    **Policy Alignment:** Update the optimal $\pi^t(y|x) \propto \exp(r(x, y; \boldsymbol{\theta}_{t,K}))$ according to (9)

**end for**

---

the data sampling step, we sample the response from the current model for the next $K$ iterations. If we take $K = 1$ and $T = \frac{\text{data size}}{\text{batch size}} * \text{epoch}$, the sampling process would be done for every iteration. In practice however we take a relative small $T$ and large $K$, because frequent on-line sampling is time consuming; see Section 4 for the implementation details.

### 3.2 Implicit Reward-learning Fine-tuning via Self-generation

So far we have seen that (4) (equivalently (5)) can efficiently utilize the demonstration dataset for better alignment. However, the computation cost for training two models (reward and policy) is significantly higher than the standard SFT. It turns out that (4) can be simplified into a supervised learning problem. Observe the following property (see Appendix B for proof):

**Lemma 3.2.** *For the loss function $\ell$ in (4), we have:*

$$\nabla_{\boldsymbol{\theta}}\ell(\boldsymbol{\theta}) = \mathbb{E}_{x\sim\rho, y\sim\pi^{\mathrm{E}}(\cdot|x), \tilde{y}\sim\pi_{\boldsymbol{\theta}}(\cdot|x)}\left[\nabla_{\boldsymbol{\theta}}\log\frac{\pi_{\boldsymbol{\theta}}(y|x)}{\pi_{\mathrm{ref}}(y|x)} - \nabla_{\boldsymbol{\theta}}\log\frac{\pi_{\boldsymbol{\theta}}(\tilde{y}|x)}{\pi_{\mathrm{ref}}(\tilde{y}|x)}\right]. \tag{6}$$

The proof of the above lemma uses the identity $r(x, y; \boldsymbol{\theta}) = \beta\log\frac{\pi_{\boldsymbol{\theta}}(y|x)}{\pi_{\mathrm{ref}}(y|x)} + \beta\log Z_{\boldsymbol{\theta}}(x)$ for some constants $Z_{\boldsymbol{\theta}}(x)$; see, e.g. Rafailov et al. [2024]. Again it is remarkable that, despite the fact that the IRL formulation only consumes the demonstration data $\mathcal{D} = x, y$, the gradient of the IRL loss takes the form as the difference of two gradients, one related to the demonstration data, the other related to the data generated by the current policy.

Lemma 3.2 leads to a simple scheme for *implicit reward-based supervised fine-tune* (IRFT) – for each training batch, it samples the response from the current model, and construct the gradient estimator (6) directly to update the parameters $\boldsymbol{\theta}$. This results in Algorithm 2, which is an SGD type algorithm for (4). In Algorithm 2, we use a double loop since generation at each step might again significantly take more time, similar to Algorithm 1. If $K = 1$ we get a single loop algorithm where we generate for every training step on the input batch.

### 3.3 Convergence Theory

We conclude the section by theoretically inspecting the proposed algorithms. Note that details and proofs of convergence theorem are moved to Appendix B due to page limits. We observe:

**Theorem 3.1.** *Under Assumption B.1, for Algorithm 1 and 2 with $\eta_t = \Theta(1/\sqrt{TK})$ we have*

$$\min_{t=1,\ldots,T,\ k=1,\ldots,K}\mathbb{E}[\|\nabla\ell(\boldsymbol{\theta}_{t,k})\|^2] \leq \mathcal{O}\left(1/\sqrt{TK} + 1/T\right).$$

Theorem 3.1 indicates that the convergence dependency is $\mathcal{O}(1/\sqrt{KT})$ (assuming $T > K$), which indicates that the algorithm could converge to stationary point if we take both the inner loop and outer loop reasonably large. This is slightly contrary to the intuition since with larger inner loop number $K$, we are having more biased estimators. This theorem shows that this biasedness actually wouldn't harm the final convergence, thus validate our practice of having a relative large inner loop number $K$ in practice (since generating at each training iteration is time-consuming).

---

**Algorithm 2:** *Implicit Reward-learning Fine-Tune* (IRFT)

---

1: **Input:** Initialize model parameter $\boldsymbol{\theta}_0(\boldsymbol{\theta}_{-1,K} = \boldsymbol{\theta}_0)$, the stepsize of reward update $\eta_t$, and $T$, $K$ the outer and inner iterations.
2: **Output:** $\hat{\boldsymbol{\theta}}$
3: **for** $t = 0, 1, ..., T-1$ **do**
4:     Take $\boldsymbol{\theta}_{t,0} = \boldsymbol{\theta}_{t-1,K}$
5:     **Data Sample:** Sample state $x_{t,k} \sim \rho$, an expert response $y_{t,k} \sim \pi^{\mathrm{E}}(\cdot|x_{t,k})$ and agent response $\tilde{y}_{t,k} \sim \pi_{\boldsymbol{\theta}_{t,0}}(\cdot|x_{t,k})$, for $k = 0, 1, ..., K-1$
6:     **for** $k = 0, 1, ..., K-1$ **do**
7:         **Estimate Gradient:** Calculate the stochastic estimator $\hat{\nabla}\ell(\boldsymbol{\theta}_{t,k})$ via (6), i.e.
$$\hat{\nabla}\ell(\boldsymbol{\theta}_{t,k}) = \nabla_{\boldsymbol{\theta}_{t,k}} \log \frac{\pi_{\boldsymbol{\theta}_{t,k}}(y_{t,k}|x_{t,k})}{\pi_{\mathrm{ref}}(y_{t,k}|x_{t,k})} - \nabla_{\boldsymbol{\theta}_{t,k}} \log \frac{\pi_{\boldsymbol{\theta}_{t,k}}(\tilde{y}_{t,k}|x_{t,k})}{\pi_{\mathrm{ref}}(\tilde{y}_{t,k}|x_{t,k})}.$$
8:         **Implicit Reward Alignment:** Update $\boldsymbol{\theta}_{t,k+1} = \boldsymbol{\theta}_{t,k} + \eta_t \hat{\nabla}\ell(\boldsymbol{\theta}_{t,k})$
9:     **end for**
10: **end for**

---

## 4 Discussions

**Implementation details of RFT**. As mentioned, training a reward model and a policy at the same time is costly. In our experiments, we discovered that the reward alignment step can be completely separated from the policy alignment step. In particular, we take $T = 1$ and $K = \frac{\text{data size}}{\text{batch size}} * \text{epoch}$ so that we train the reward over the entire dataset and then switch to the policy alignment. In our experiments, we indeed observe that only one round of above procedure can readily show superior performance over SFT and implicit reward-learning methods for `pythia-1.4b` model.

**Implementation details of IRFT**. It is worth noticing that in (6), the policy $\pi$ is not parameterized by $\boldsymbol{\theta}$ directly. In our numerical experiment, we directly parameterize the LLM $\pi$ by $\boldsymbol{\theta}$, making (6) the gradient of an supervised optimization problem itself. Meanwhile, it is not straightforward to calculate the self-generation gradient (6) directly, thus we need to design a loss function for back-propagation in main-stream packages such as PyTorch and TensorFlow. In practice, at each training iteration we first sample $\tilde{y} \sim \pi(\cdot|x; \boldsymbol{\theta})$ and pass the following loss function

$$h\left(\log \frac{\pi(y|x; \boldsymbol{\theta})}{\pi_{\mathrm{ref}}(y|x)} - \log \frac{\pi(\tilde{y}|x; \boldsymbol{\theta})}{\pi_{\mathrm{ref}}(\tilde{y}|x)}\right) \tag{7}$$

into the standard optimizers (such as `SGD` or `Adam`) for back-propagation. Here $h$ is a nonlinear function. We take $h = \log \sigma$ where $\sigma$ is the logistic loss function $\sigma(t) := \log(1 + \exp(-t))$ as in Rafailov et al. [2024], Chen et al. [2024] for its non-negativity, smoothness and exponentially decaying tail to avoid excessive growth in the absolute value of the log-likelihood.

**Discussion on the computational costs**. For Algorithm 1, we need to maintain a reward model and a policy model (which is the LLM), and this is doubling the standard LLM fine-tuning. Thus the memory consumption and computation time of Algorithm 1 is similar to the standard RLHF process (RLHF = reward learning + policy optimization); For Algorithm 2, we simply need to maintain the policy (LLM) model, and the memory consumption would be exactly the same as the standard SFT, whereas the computation time would involving generating for the entire training sample, which would be of similar level as the standard policy optimization process (same computational time as SPIN). Note that standard policy optimization process is equivalent to the time of standard SFT and a generation process toward all training input prompts.

We summarize the memory consumption and the computational time of the proposed methods in Table 2, assuming that the reward and policy models are of same size. Here "Forward" means the memory required for storing a model in inference mode, and "Backward" is the memory required for storing a model in training mode,

| Method | Peak Memory | Computation Time |
|---|---|---|
| Algorithm 1 | Forward+Backward | 2SFT+Generation |
| Algorithm 2 | Backward | SFT+Generation |

Table 2: Table summarizing the computational costs of proposed methods.

including weights, activations and gradients; also "SFT" means the computational time as standard

SFT, and "Generation" means the time to generate continuations for each of the input training prompts. Therefore "2SFT+Generation" is roughly the same time as standard RLHF.

**Comparison to SPIN**. We discuss here the connection between our proposed algorithms with the self-play fine-tune algorithm (SPIN in Chen et al. [2024]), which also maximizes the gap between two rewards. First, SPIN is motivated by certain two-player games, while in our case, we show that the difference of two rewards in (5) naturally comes from *a single*, *reward learning* agent; see (4).

Second, IRFT covers SPIN as a special case. In particular, if we take $T = 1$ and $K$ as the total number of training iterations, the IRFT algorithm is equivalent to SPIN. In practice, we tested on different choices of $T$ and show that a reasonable generation frequency can results in a strong model performance.

Finally, since SPIN does not involve explicit reward learning, its connection to RFT is relatively remote. It is worth noting that the relation between the proposed Algorithm 1 and Algorithm 2 is similar to that of RLHF to DPO. There has been intensive discussions regarding whether reward-based or reward-free algorithm gives better model performances, but this topic is beyond the scope of the current paper. We refer to Xu et al. [2024] for a comprehensive study.

## 5   Numerical experiments

In this section we study the proposed Algorithm 1 and 2 numerically. Our experiment mainly show the advantages of the proposed methods in the following aspects: (1) Reward learning is key to improve over standard SFT, even if we do not have preference dataset; (2) The double loop design in both Algorithm 1 and 2 enable us to explore appropriate parameter settings that could break the performance limits of the state-of-the-art methods, including SFT and SPIN.

### 5.1   Experiment Setup

**Model and Datasets.** Since reward-based methods can be costly by training two models at the same time, we mainly test Algorithm 1 on `pythia-1b` reward model and `pythia-1.4b` policy model [Biderman et al., 2023]. We tested pythia on Anthropic-HH dataset [Bai et al., 2022]. Anthropic-HH is a preference dataset that provide two continuations based on helpfulness and harmlessness, and we only pick 10k chosen/preferred continuation data to form the demonstration dataset, which enable us to check the log likelihood of the non-preferred continuation without feeding the model with such data. At each iteration, we train our model for 2 epochs (seeing each data for two times).

Algorithm 2 is tested on two models: `pythia-1.4b` and `zephyr-7b-sft-full` [Tunstall et al., 2023]. We tested on Ultrachat200k dataset by HuggingFace, which is a subset of the high quality demonstration UltraChat dataset[Ding et al., 2023] for text generation and dialogue. For Ultrachat200k, we adopt the same strategy as Chen et al. [2024] to pick up 50k data for training. At each iteration, we again train our model for 2 epochs.

**Evaluation.**     For the Anthropic-HH dataset, we show the reward evaluated by the `PKU-Alignment/beaver-7b-v3.0-reward` [Dai et al., 2024, Ji et al., 2023] model which is a popular 7b model fine-tuned from `meta-llama/Llama-2-7b` tailored for evaluating human preferences regarding helpfulness and harmlessness. We also record win rate of the two proposed methods over base model and SFT model. For the Ultrachat200k dataset, we follow the widely used Hugging-Face Open LLM Leaderboard [Beeching et al., 2023]. This evaluation package assess an LLM based on six tasks: LLMs on commonsense reasoning (Arc Clark et al. [2018], HellaSwag Zellers et al. [2019], Winogrande Sakaguchi et al. [2021]), multi-task language understanding (MMLU Hendrycks et al. [2020]), human falsehood mimic (TruthfulQA Lin et al. [2021]) and math problem solving (GSM8K, Cobbe et al. [2021]). See the appendix for more implementation details.

### 5.2   Results of RFT (Algorithm 1)

We present the result of Algorithm 1 over Anthropic-HH dataset. We first fine-tuned `pythia-1.4b` using supervised fine-tune over the entire dataset (160k training data in total) using only the preferred/chosen data for 10 epochs and pick up the checkpoint with the best testing accuracy as our base model. We then use `PKU-Alignment/beaver-7b-v3.0-reward` model as our ground truth reward model. We use this model to pick 10k data from Anthropic-HH dataset with the highest reward scores. Next, we fine-tune the base model using SFT and Algorithm 1. Figure 2 shows the experiment results on averaged reward and win rate, where we record the average score (by

`PKU-Alignment/beaver-7b-v3.0-reward`) of the continuation generated for test datasets, also the win rate (ratio of samples where the reward of our model's generation is higher than the model compared) of the proposed Algorithm 1 over the full SFT base model and the top 10k SFT model. The figures show that the proposed algorithm improves over the SFT models in terms of effectively improve the helpfulness and harmlessness of the model continuation.

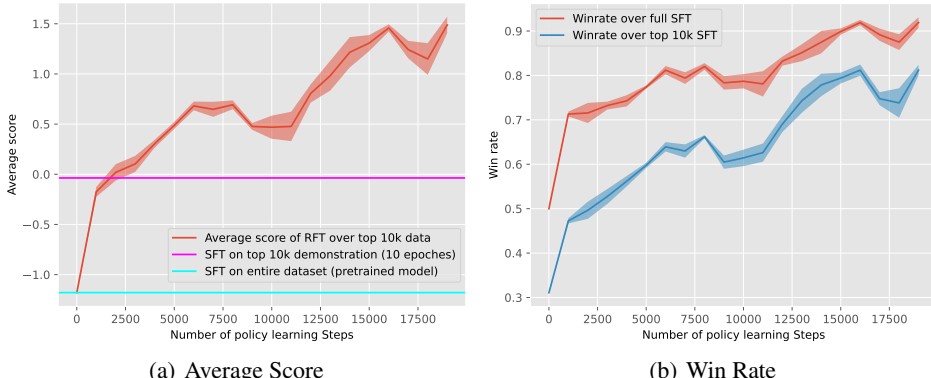

(a) Average Score          (b) Win Rate

Figure 2: Algorithm 1 fine-tuning result of `pythia-1.4b` over Anthropic-HH (with top 10k data picked by `PKU-Alignment/beaver-7b-v3.0-reward`). We record the average score of test dataset on the left figure and the win rate of Algorithm 1 over the (full SFT) base model and the SFT model.

We remind the readers that the advantage of Algorithm 1 over SFT in Figure 2 can be partially explained by Figure 1 right side: despite the fact that SFT, Algorithm 1 and 2 are only observing chosen/preferred data, the latter two still outperforms SFT since they discourage the likelihood of the synthetic non-preferred data, thus bringing better performance and robustness for the model.

## 5.3 Results of IRFT (Algorithm 2)

Different from the time consuming Algorithm 1, Algorithm 2 is more capable of handling large data and models. We first present the result for `pythia-1.4b models` over Ultrachat200k data. We remind the reader again that $T = 1$ in Algorithm 2 is equivalent to SPIN [Chen et al., 2024][1]. We tested on different choices of $T$ and identify that $T = 5$ to 8 gives the best performance in the Open LLM Leaderboard evaluations.

The Open LLM Leaderboard result is presented in Table 3. We have the following main observations based on the results in Table 3:

1. SFT is not efficient in terms of boosting the pre-trained model performance on downstream tasks comparing to methods which promote the decreasing of the likelihood of synthetic data, namely SPIN and IRFT;

2. SPIN and IRFT (Algorithm 2) are both capable of further improving the performance of pythia model over downstream tasks, whereas IRFT shows better results due to more frequent generation comparing to SPIN. IRFT with $T > 1$ outperforms both SFT and SPIN on most of the tasks as well as the average score;

3. More frequent generation might also result in more variances, therefore a reasonable $T$ (around 5) results in the best evaluation performance. Careful hyperparameter tuning might be needed for different models and datasets when applying our method, while we recommend using $T = 5$ as the default setting.

Apparently 1b model is not strong enough to handle hard tasks, e.g. GSM8k and all model performances are not desirable. Now we present the result for `zephyr-7b-sft-full`. We remind the reader that this is a fully SFT-ed model and further SFT would only detriment the model performance (see Chen et al. [2024]). The results are presented in Table 4 where we can see that similar to the 1b case, both SPIN and IRFT could effectively improve the performance of SFT-ed model and the average performance of IRFT with $T = 5$ stands out. The success of IRFT and SPIN further suggest that reward learning is indeed beneficial for aligning with demonstration data.

---

[1]IRFT with $T = 1$ and 2 epochs is equivalent to SPIN iteration 0, and $T = 2$ with 2 epochs for each $T$ is equivalent to SPIN iteration 1, etc.

Table 3: Test performance of SPIN [Chen et al., 2024] and IRFT (Algorithm 2) based on `pythia-1.4b` across HuggingFace Open LLM Leaderboard datasets. We keep training for 2 epochs after each generation process and $K$ are calculated after this rule.

| Tasks Metrics | $T$ | $K$ | AI2_Arc acc_norm | TruthfulQA acc | Winogrande acc | GSM8k exact_match | HellaSwag acc_norm | MMLU acc | Average |
|---|---|---|---|---|---|---|---|---|---|
| `pythia-1.4b` | 0 | 0 | 54.54 | 31.00 | 57.46 | 1.44 | 53.55 | 25.63 | 37.27 |
| SFT | 0 | $\frac{\#\text{ samples}}{\text{batchsize}} * 2$ | 54.74 | 30.93 | 57.30 | 2.05 | 52.98 | 25.62 | 37.27 |
| IRFT (SPIN) | 1 | $\frac{\#\text{ samples}}{\text{batchsize}} * 2$ | 54.00 | 31.73 | 57.70 | 1.36 | 53.76 | 25.54 | 37.35 |
| IRFT (SPIN iter 2) | 2 | $\frac{\#\text{ samples}}{\text{batchsize}} * 2$ | 52.85 | **32.04** | 57.38 | 1.74 | 53.57 | 25.49 | 37.18 |
| IRFT | 5 | $\frac{\#\text{ samples}}{\text{batchsize}} * \frac{2}{5}$ | 53.75 | 31.67 | 56.91 | 1.74 | **54.79** | 25.32 | 37.36 |
| IRFT | 10 | $\frac{\#\text{ samples}}{\text{batchsize}} * \frac{2}{5}$ | 53.75 | 31.92 | 57.85 | **2.43** | 54.77 | 25.44 | **37.69** |
| IRFT | 8 | $\frac{\#\text{ samples}}{\text{batchsize}} * \frac{2}{8}$ | 53.75 | 31.40 | 56.91 | 2.35 | 54.62 | 25.52 | 37.43 |
| IRFT | 16 | $\frac{\#\text{ samples}}{\text{batchsize}} * \frac{2}{8}$ | **56.34** | 31.54 | **58.41** | 1.59 | 54.54 | **25.69** | 37.57 |

Table 4: Test performance of SPIN [Chen et al., 2024] and IRFT (Algorithm 2) based on `zephyr-7b-sft-full` across HuggingFace Open LLM Leaderboard datasets.

| Tasks Metrics | T | K | AI2_Arc acc_norm | TruthfulQA acc | Winogrande acc | GSM8k exact_match | HellaSwag acc_norm | MMLU acc | Average |
|---|---|---|---|---|---|---|---|---|---|
| `zephyr-7b-sft-full` | 0 | 0 | 74.83 | 34.07 | 76.09 | 31.92 | 81.09 | **58.86** | 59.48 |
| IRFT (SPIN) | 1 | $\frac{\#\text{ samples}}{\text{batchsize}} * 2$ | 75.08 | 36.57 | 76.01 | 33.59 | 82.81 | 57.83 | 60.32 |
| IRFT (SPIN iter 2) | 2 | $\frac{\#\text{ samples}}{\text{batchsize}} * 2$ | 76.13 | 36.56 | 76.64 | **35.56** | **83.39** | 57.82 | 61.02 |
| IRFT | 5 | $\frac{\#\text{ samples}}{\text{batchsize}} * \frac{2}{5}$ | 75.82 | **39.99** | 77.19 | 31.24 | 82.07 | 57.93 | 60.71 |
| IRFT | 10 | $\frac{\#\text{ samples}}{\text{batchsize}} * \frac{2}{5}$ | **76.78** | 36.84 | **77.43** | 34.34 | 83.05 | 57.72 | **61.03** |
| IRFT | 8 | $\frac{\#\text{ samples}}{\text{batchsize}} * \frac{2}{8}$ | 75.23 | 36.67 | 75.85 | 31.84 | 80.89 | 58.60 | 59.85 |
| IRFT | 16 | $\frac{\#\text{ samples}}{\text{batchsize}} * \frac{2}{8}$ | 75.79 | 35.55 | 76.56 | 32.52 | 82.3 | 58.77 | 60.25 |

# 6 Conclusions and Limitations

In this paper we proposed reward-learning approaches for aligning LLMs with demonstration datasets. We show both theoretically and numerically the great potential of reward-learning for alignment even without preference dataset. Our theory only indicate the convergence of the proposed algorithm to stationary point, and it is not clear what the policy converges to. The additional computation resources required for tuning two models or generate synthetic data in our algorithms are not negligible. Future works include exploring reward-learning for larger models and more complicated demonstration tasks, boosting the algorithm efficiency, and understanding how synthetic negative sample helps the LLMs to distinguish the preference dataset, etc.

## Acknowledgments

M. Hong, S. Zeng and J. Li are supported partially by NSF under the grants EPCN-2311007, ECCS-2426064 and CCF-2414372, also by Minnesota Supercomputing Institute. A. Garcia and C. Li are partially supported by ECCS-2240789 and CCF-2414373.

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

# Appendix

## A Related works

Fine-tuning language models is prevailing to improve LLMs performance on various instructional tasks, and has shown great success in enabling LLMs to generalize to efficiently respond out-of-sample instructions [Chung et al., 2024]. Despite many successful applications of SFT, people soon realized the great potential of reward learning and reinforcement learning based fine-tuning over preference datasets for different tasks, including text-summarizing [Liu et al., 2020, Ziegler et al., 2019], story-telling [Ziegler et al., 2019], instruction-following [Ouyang et al., 2022, Ramamurthy et al., 2022], etc. Equipped with the popular Bradley-Terry model [Bradley and Terry, 1952], RLHF fine-tune a language model using policy optimization methods, such as REINFORCE [Williams, 1992], proximal policy optimization (PPO, Schulman et al. [2017]) and a lot more. On major obstacle for preference dataset fine-tuning is the costly and time-consuming process of human labeling, and methods such as self-play fine-tune (SPIN, Chen et al. [2024]), synthetic data with binary feedback in self-training [Singh et al., 2023], weak-to-strong generalization [Burns et al., 2023] and self-rewarding fine-tuning [Yuan et al., 2024] seek for improvement over SFT under weaker data supervisions comparing to preference datasets. In particular, SPIN generates synthetic samples for input prompts in the demonstration dataset and use them as the rejected data to for a 'pseudo' preference data. As we will see, SPIN actually coincides with our implicit reward learning approach where we motivate the synthetic data in a more natural way.

In the reinforcement learning (RL) literature, inverse reinforcement learning (IRL) proposes to jointly learn the reward $r$ which best explains an expert policy $\pi^E$ and the policy $\pi$ which in turn mimics this expert policy $\pi^E$ from demonstration data. The most popular framework is the maximum entropy IRL (MaxEnt-IRL) framework Ziebart et al. [2008], Levine et al. [2011], Ziebart et al. [2013], Bloem and Bambos [2014], Zeng et al. [2022], which seeks for a policy maximizing the entropic-regularized reward that matches the empirical averages in expert's demonstrations data. MaxEnt-IRL utilizes only the demonstration dataset for reward learning and already yields superior performance over the plain behavior cloning [Pomerleau, 1988, Osa et al., 2018] approach on various RL tasks.

## B Proofs for Section 3

We restate and prove Lemma 3.1:

**Lemma B.1.** *Problem* (4) *is equivalent to the following minimax optimization problem:*

$$\max_{\boldsymbol{\theta}} \min_{\pi} \mathbb{E}_{x\sim\rho, y\sim\pi^{\mathrm{E}}(\cdot|x), \tilde{y}\sim\pi(\cdot|x)} \left[ \frac{r(x, y; \boldsymbol{\theta}) - r(x, \tilde{y}; \boldsymbol{\theta})}{\beta} + D_{\mathrm{KL}}\Big(\pi(\cdot|x)\|\pi_{ref}(\cdot|x)\Big) \right]. \quad (8)$$

**Proof.** It is straightforward to see that the lower-level problem in (4) enjoys a closed-form solution:

$$\pi_{\boldsymbol{\theta}}(y|x) = \frac{\pi_{\mathrm{ref}}(y|x) \exp\left(\frac{1}{\beta} r(x, y; \boldsymbol{\theta})\right)}{\sum_{\tilde{y}\in\mathcal{A}} \pi_{\mathrm{ref}}(\tilde{y}|x) \exp\left(\frac{1}{\beta} r(x, \tilde{y}; \boldsymbol{\theta})\right)} \quad (9)$$

where $\mathcal{A}$ is the set of all possible responses. Plugging (9) into (4), we obtain:

$$\max_{\boldsymbol{\theta}} \ \mathbb{E}_{x\sim\rho, y\sim\pi^{\mathrm{E}}(\cdot|x)} \left[ \log\left(\pi_{\mathrm{ref}}(y|x) \exp\left(\frac{1}{\beta} r(x, y; \boldsymbol{\theta})\right)\right) - \log\left(\sum_{\tilde{y}\in\mathcal{A}} \pi_{\mathrm{ref}}(\tilde{y}|x) \exp\left(\frac{1}{\beta} r(x, \tilde{y}; \boldsymbol{\theta})\right)\right) \right] \quad (10)$$

Utilizing the following identity:

$$\log\left(\sum_{\tilde{y}\in\mathcal{A}} \pi_{\mathrm{ref}}(\tilde{y}|x) \exp\left(\frac{1}{\beta} r(x, \tilde{y}; \boldsymbol{\theta})\right)\right) = \max_{\pi} \mathbb{E}_{\tilde{y}\sim\pi(\cdot|x)}[\frac{1}{\beta} r(x, \tilde{y}; \boldsymbol{\theta})] - D_{\mathrm{KL}}\Big(\pi(\cdot|x)\|\pi_{\mathrm{ref}}(\cdot|x)\Big)$$

we obtain the following max-min problem (omitting some constant terms):

$$\max_{\boldsymbol{\theta}} \min_{\pi} \mathbb{E}_{x\sim\rho, y\sim\pi^{\mathrm{E}}(\cdot|x), \tilde{y}\sim\pi(\cdot|x)} \left[ \frac{r(x, y; \boldsymbol{\theta}) - r(x, \tilde{y}; \boldsymbol{\theta})}{\beta} + D_{\mathrm{KL}}\Big(\pi(\cdot|x)\|\pi_{\mathrm{ref}}(\cdot|x)\Big) \right] \quad (11)$$

The proof is completed. □

Next, we restate and prove Lemma 3.2:

**Lemma B.2.** *For the loss function $\ell$ in (4), we have:*

$$\nabla_{\boldsymbol{\theta}}\ell(\boldsymbol{\theta}) = \mathbb{E}_{x\sim\rho,y\sim\pi^{\mathrm{E}}(\cdot|x),\tilde{y}\sim\pi_{\boldsymbol{\theta}}(\cdot|x)}\left[\nabla_{\boldsymbol{\theta}}\log\frac{\pi_{\boldsymbol{\theta}}(y|x)}{\pi_{\mathrm{ref}}(y|x)} - \nabla_{\boldsymbol{\theta}}\log\frac{\pi_{\boldsymbol{\theta}}(\tilde{y}|x)}{\pi_{\mathrm{ref}}(\tilde{y}|x)}\right] \tag{12}$$

*which we refer to as the **self-generation gradient**, since at each iteration one need to generate one sample output $\tilde{y}$ from the current policy $\pi_{\boldsymbol{\theta}}$ and calculate the difference of the two rewards.*

**Proof.** Omitting the constant terms not related to $\boldsymbol{\theta}$ in (10), we have

$$\max_{\boldsymbol{\theta}}\ \ell(\boldsymbol{\theta}) = \mathbb{E}_{x\sim\rho,y\sim\pi^{\mathrm{E}}(\cdot|x)}\left[\frac{1}{\beta}r(x,y;\boldsymbol{\theta}) - \log\left(\sum_{\tilde{y}\in\mathcal{A}}\pi_{\mathrm{ref}}(\tilde{y}|x)\exp\left(\frac{1}{\beta}r(x,\tilde{y};\boldsymbol{\theta})\right)\right)\right] \tag{13}$$

Calculating the derivative we get

$$\nabla_{\boldsymbol{\theta}}\ell(\boldsymbol{\theta}) = \frac{1}{\beta}\mathbb{E}_{x\sim\rho,y\sim\pi^{\mathrm{E}}(\cdot|s)}[\nabla_{\boldsymbol{\theta}}r(x,y;\boldsymbol{\theta})] - \mathbb{E}_{x\sim\rho}\left[\nabla_{\boldsymbol{\theta}}\log\left(\sum_{\tilde{y}\in\mathcal{A}}\pi_{\mathrm{ref}}(\tilde{y}|x)\exp\left(\frac{1}{\beta}r(x,\tilde{y};\boldsymbol{\theta})\right)\right)\right]$$

$$= \frac{1}{\beta}\mathbb{E}_{x\sim\rho,y\sim\pi^{\mathrm{E}}(\cdot|x)}[\nabla_{\boldsymbol{\theta}}r(x,y;\boldsymbol{\theta})] - \frac{1}{\beta}\mathbb{E}_{x\sim\rho}\left[\sum_{y\in\mathcal{A}}\frac{\pi_{\mathrm{ref}}(y|x)\exp\left(\frac{1}{\beta}r(x,y;\boldsymbol{\theta})\right)}{\sum_{\tilde{y}\in\mathcal{A}}\pi_{\mathrm{ref}}(\tilde{y}|x)\exp\left(\frac{1}{\beta}r(x,\tilde{y};\boldsymbol{\theta})\right)}\nabla_{\boldsymbol{\theta}}r(x,y;\boldsymbol{\theta})\right]$$

$$= \frac{1}{\beta}\mathbb{E}_{x\sim\rho,y\sim\pi^{\mathrm{E}}(\cdot|x)}[\nabla_{\boldsymbol{\theta}}r(x,y;\boldsymbol{\theta})] - \frac{1}{\beta}\mathbb{E}_{x\sim\rho,y\sim\pi_{\boldsymbol{\theta}}(\cdot|s)}[\nabla_{\boldsymbol{\theta}}r(x,y;\boldsymbol{\theta})]$$

$$= \frac{1}{\beta}\mathbb{E}_{x\sim\rho,y\sim\pi^{\mathrm{E}}(\cdot|x),\tilde{y}\sim\pi_{\boldsymbol{\theta}}(\cdot|x)}[\nabla_{\boldsymbol{\theta}}r(x,y;\boldsymbol{\theta}) - \nabla_{\boldsymbol{\theta}}r(x,\tilde{y};\boldsymbol{\theta})]$$

which implies that to minimize $\ell(\boldsymbol{\theta})$, one should always generate samples based on the current estimation of the policy $\tilde{y}\sim\pi_{\boldsymbol{\theta}}(\cdot|x)$ and then update.

Now from (9) we get:

$$r(x,y;\boldsymbol{\theta}) = \beta\log\frac{\pi_{\boldsymbol{\theta}}(y|x)}{\pi_{\mathrm{ref}}(y|x)} + \beta\log Z_{\boldsymbol{\theta}}(x) \tag{14}$$

where $Z_{\boldsymbol{\theta}}(x)$ is the denominator of (9). In the view of (14), we can actually directly estimate:

$$\nabla_{\boldsymbol{\theta}}\ell(\boldsymbol{\theta}) = \mathbb{E}_{x\sim\rho,y\sim\pi^{\mathrm{E}}(\cdot|x),\tilde{y}\sim\pi_{\boldsymbol{\theta}}(\cdot|x)}\left[\nabla_{\boldsymbol{\theta}}\log\frac{\pi_{\boldsymbol{\theta}}(y|x)}{\pi_{\mathrm{ref}}(y|x)} - \nabla_{\boldsymbol{\theta}}\log\frac{\pi_{\boldsymbol{\theta}}(\tilde{y}|x)}{\pi_{\mathrm{ref}}(\tilde{y}|x)}\right] \tag{15}$$

The proof is completed. □

Now we move to the proof for Section 3.3. We state the assumption needed for proving the final result:

**Assumption B.1.** *For Algorithm 1 and 2, we assume that*

1. *The policy distribution $\pi_{\boldsymbol{\theta}}$ is uniformly lower and upper bounded, i.e.*

$$\pi_{\min} \le \|\pi_{\boldsymbol{\theta}}(\cdot|x)\|_{\infty} \le \pi_{\max}$$

   *where $0 < \pi_{\min} < \pi_{\max}$, for all $x$;*

2. *$\nabla\pi_{\boldsymbol{\theta}}$ is bounded, i.e. $\|\nabla\pi_{\boldsymbol{\theta}}(\cdot|x)\| \le L_0$ for all $x$;*

3. *$\nabla\pi_{\boldsymbol{\theta}}$ is Lipschitz, i.e. $\|\nabla\pi_{\boldsymbol{\theta}_1}(y|x) - \nabla\pi_{\boldsymbol{\theta}_2}(y|x)\| \le L_1\|\boldsymbol{\theta}_1 - \boldsymbol{\theta}_2\|$, for all $x$ and $y$;*

*where $\pi_{\boldsymbol{\theta}}$ is as defined in (9).*

The above assumption can readily establish the assumption below, which is needed for our final convergence result.

**Assumption B.2.** *For Algorithm 2, we assume that*

*1. $\ell$ is L-Lipschitz smooth w.r.t. $\boldsymbol{\theta}$, i.e.*

$$\|\nabla\ell(\boldsymbol{\theta}_1) - \nabla\ell(\boldsymbol{\theta}_2)\| \leq L\|\boldsymbol{\theta}_1 - \boldsymbol{\theta}_2\|$$

*2. The stochastic estimator $\hat{\nabla}\ell$ is bounded, i.e.*

$$\|\hat{\nabla}\ell(\boldsymbol{\theta})\| \leq G$$

These are all standard assumptions in nonconvex smooth stochastic optimization. We have the following lemma:

**Lemma B.3.** *If Assumption B.1 holds, Assumption B.2 also holds with the following parameters:*

$$L = \frac{L_0(3L_0 + L_1)}{\pi_{\min}^2}, \ G = \frac{2L_0}{\pi_{\min}}$$

**Proof.** We just show the value for $L$ since $G$ can be similarly computed. Since

$$\nabla\ell(\boldsymbol{\theta}) = \frac{1}{\beta}\mathbb{E}_{x\sim\rho, y\sim\pi^{\mathrm{E}}(\cdot|x), \tilde{y}\sim\pi_{\boldsymbol{\theta}}(\cdot|x)}\left[\nabla_{\boldsymbol{\theta}}\log\frac{\pi_{\boldsymbol{\theta}}(y|x)}{\pi_{\mathrm{ref}}(y|x)} - \nabla_{\boldsymbol{\theta}}\log\frac{\pi_{\boldsymbol{\theta}}(\tilde{y}|x)}{\pi_{\mathrm{ref}}(\tilde{y}|x)}\right]$$

we have

$$
\begin{aligned}
&\|\nabla\ell(\boldsymbol{\theta}_1) - \nabla\ell(\boldsymbol{\theta}_2)\| \\
=&\frac{1}{\beta}\left\|\mathbb{E}_{x\sim\rho, y\sim\pi^{\mathrm{E}}(\cdot|x), \tilde{y}\sim\pi_{\boldsymbol{\theta}_1}(\cdot|x)}\left[\nabla_{\boldsymbol{\theta}}\log\frac{\pi_{\boldsymbol{\theta}_1}(y|x)}{\pi_{\mathrm{ref}}(y|x)} - \nabla_{\boldsymbol{\theta}}\log\frac{\pi_{\boldsymbol{\theta}_1}(\tilde{y}|x)}{\pi_{\mathrm{ref}}(\tilde{y}|x)}\right]\right.\\
&\left. - \mathbb{E}_{x\sim\rho, y\sim\pi^{\mathrm{E}}(\cdot|x), \tilde{y}\sim\pi_{\boldsymbol{\theta}_2}(\cdot|x)}\left[\nabla_{\boldsymbol{\theta}}\log\frac{\pi_{\boldsymbol{\theta}_2}(y|x)}{\pi_{\mathrm{ref}}(y|x)} - \nabla_{\boldsymbol{\theta}}\log\frac{\pi_{\boldsymbol{\theta}_2}(\tilde{y}|x)}{\pi_{\mathrm{ref}}(\tilde{y}|x)}\right]\right\| \\
\leq&\frac{1}{\beta}\left\|\mathbb{E}_{x\sim\rho, y\sim\pi^{\mathrm{E}}(\cdot|x), \tilde{y}\sim\pi_{\boldsymbol{\theta}_1}(\cdot|x)}\left[\nabla_{\boldsymbol{\theta}}\log\frac{\pi_{\boldsymbol{\theta}_1}(y|x)}{\pi_{\mathrm{ref}}(y|x)} - \nabla_{\boldsymbol{\theta}}\log\frac{\pi_{\boldsymbol{\theta}_1}(\tilde{y}|x)}{\pi_{\mathrm{ref}}(\tilde{y}|x)}\right]\right.\\
&\left. - \mathbb{E}_{x\sim\rho, y\sim\pi^{\mathrm{E}}(\cdot|x), \tilde{y}\sim\pi_{\boldsymbol{\theta}_1}(\cdot|x)}\left[\nabla_{\boldsymbol{\theta}}\log\frac{\pi_{\boldsymbol{\theta}_2}(y|x)}{\pi_{\mathrm{ref}}(y|x)} - \nabla_{\boldsymbol{\theta}}\log\frac{\pi_{\boldsymbol{\theta}_2}(\tilde{y}|x)}{\pi_{\mathrm{ref}}(\tilde{y}|x)}\right]\right\| \\
&+ \frac{1}{\beta}\left\|\mathbb{E}_{x\sim\rho, y\sim\pi^{\mathrm{E}}(\cdot|x), \tilde{y}\sim\pi_{\boldsymbol{\theta}_1}(\cdot|x)}\left[\nabla_{\boldsymbol{\theta}}\log\frac{\pi_{\boldsymbol{\theta}_2}(y|x)}{\pi_{\mathrm{ref}}(y|x)} - \nabla_{\boldsymbol{\theta}}\log\frac{\pi_{\boldsymbol{\theta}_2}(\tilde{y}|x)}{\pi_{\mathrm{ref}}(\tilde{y}|x)}\right]\right.\\
&\left. - \mathbb{E}_{x\sim\rho, y\sim\pi^{\mathrm{E}}(\cdot|x), \tilde{y}\sim\pi_{\boldsymbol{\theta}_2}(\cdot|x)}\left[\nabla_{\boldsymbol{\theta}}\log\frac{\pi_{\boldsymbol{\theta}_2}(y|x)}{\pi_{\mathrm{ref}}(y|x)} - \nabla_{\boldsymbol{\theta}}\log\frac{\pi_{\boldsymbol{\theta}_2}(\tilde{y}|x)}{\pi_{\mathrm{ref}}(\tilde{y}|x)}\right]\right\|
\end{aligned}
\tag{16}
$$

For the first part, since

$$\nabla\log\pi_{\boldsymbol{\theta}}(y|x) = \frac{\nabla\pi_{\boldsymbol{\theta}}(y|x)}{\pi_{\boldsymbol{\theta}}(y|x)}$$

we have

$$
\begin{aligned}
&\frac{1}{\beta}\left\|\mathbb{E}_{x\sim\rho, y\sim\pi^{\mathrm{E}}(\cdot|x), \tilde{y}\sim\pi_{\boldsymbol{\theta}_1}(\cdot|x)}\left[\nabla_{\boldsymbol{\theta}}\log\frac{\pi_{\boldsymbol{\theta}_1}(y|x)}{\pi_{\mathrm{ref}}(y|x)} - \nabla_{\boldsymbol{\theta}}\log\frac{\pi_{\boldsymbol{\theta}_1}(\tilde{y}|x)}{\pi_{\mathrm{ref}}(\tilde{y}|x)} - \nabla_{\boldsymbol{\theta}}\log\frac{\pi_{\boldsymbol{\theta}_2}(y|x)}{\pi_{\mathrm{ref}}(y|x)} + \nabla_{\boldsymbol{\theta}}\log\frac{\pi_{\boldsymbol{\theta}_2}(\tilde{y}|x)}{\pi_{\mathrm{ref}}(\tilde{y}|x)}\right]\right\| \\
\leq&\frac{1}{\beta}\mathbb{E}_{x\sim\rho, y\sim\pi^{\mathrm{E}}(\cdot|x), \tilde{y}\sim\pi_{\boldsymbol{\theta}_1}(\cdot|x)}\left\|\frac{\nabla\pi_{\boldsymbol{\theta}_1}(y|x)}{\pi_{\boldsymbol{\theta}_1}(y|x)} - \frac{\nabla\pi_{\boldsymbol{\theta}_2}(y|x)}{\pi_{\boldsymbol{\theta}_2}(y|x)} - \frac{\nabla\pi_{\boldsymbol{\theta}_1}(\tilde{y}|x)}{\pi_{\boldsymbol{\theta}_1}(\tilde{y}|x)} + \frac{\nabla\pi_{\boldsymbol{\theta}_2}(\tilde{y}|x)}{\pi_{\boldsymbol{\theta}_2}(\tilde{y}|x)}\right\| \\
\leq&\frac{1}{\beta}\mathbb{E}_{x\sim\rho, y\sim\pi^{\mathrm{E}}(\cdot|x), \tilde{y}\sim\pi_{\boldsymbol{\theta}_1}(\cdot|x)}\left\|\frac{\nabla\pi_{\boldsymbol{\theta}_1}(y|x)}{\pi_{\boldsymbol{\theta}_1}(y|x)} - \frac{\nabla\pi_{\boldsymbol{\theta}_2}(y|x)}{\pi_{\boldsymbol{\theta}_2}(y|x)}\right\| + \frac{1}{\beta}\mathbb{E}_{x\sim\rho, y\sim\pi^{\mathrm{E}}(\cdot|x), \tilde{y}\sim\pi_{\boldsymbol{\theta}_1}(\cdot|x)}\left\|\frac{\nabla\pi_{\boldsymbol{\theta}_1}(\tilde{y}|x)}{\pi_{\boldsymbol{\theta}_1}(\tilde{y}|x)} - \frac{\nabla\pi_{\boldsymbol{\theta}_2}(\tilde{y}|x)}{\pi_{\boldsymbol{\theta}_2}(\tilde{y}|x)}\right\| \\
\leq&\frac{1}{\beta}\mathbb{E}_{x\sim\rho, y\sim\pi^{\mathrm{E}}(\cdot|x), \tilde{y}\sim\pi_{\boldsymbol{\theta}_1}(\cdot|x)}\frac{\|\pi_{\boldsymbol{\theta}_2}(y|x)\nabla\pi_{\boldsymbol{\theta}_1}(y|x) - \pi_{\boldsymbol{\theta}_1}(y|x)\nabla\pi_{\boldsymbol{\theta}_2}(y|x)\|}{\pi_{\boldsymbol{\theta}_1}(y|x)\pi_{\boldsymbol{\theta}_2}(y|x)} + \text{(same term for } \tilde{y}) \\
\leq&\frac{2}{\beta}\frac{\pi_{\max}L_1 + L_0^2}{\pi_{\min}^2}\|\boldsymbol{\theta}_1 - \boldsymbol{\theta}_2\|
\end{aligned}
$$

For the second term in the last line of (16), we have

$$
\begin{aligned}
&\frac{1}{\beta}\left\|\mathbb{E}_{x\sim\rho,y\sim\pi^{\mathrm{E}}(\cdot|x),\tilde{y}\sim\pi_{\boldsymbol{\theta}_1}(\cdot|x)}\left[\nabla_{\boldsymbol{\theta}}\log\frac{\pi_{\boldsymbol{\theta}_2}(y|x)}{\pi_{\mathrm{ref}}(y|x)}-\nabla_{\boldsymbol{\theta}}\log\frac{\pi_{\boldsymbol{\theta}_2}(\tilde{y}|x)}{\pi_{\mathrm{ref}}(\tilde{y}|x)}\right]\right.\\
&\quad\left.-\mathbb{E}_{x\sim\rho,y\sim\pi^{\mathrm{E}}(\cdot|x),\tilde{y}\sim\pi_{\boldsymbol{\theta}_2}(\cdot|x)}\left[\nabla_{\boldsymbol{\theta}}\log\frac{\pi_{\boldsymbol{\theta}_2}(y|x)}{\pi_{\mathrm{ref}}(y|x)}-\nabla_{\boldsymbol{\theta}}\log\frac{\pi_{\boldsymbol{\theta}_2}(\tilde{y}|x)}{\pi_{\mathrm{ref}}(\tilde{y}|x)}\right]\right\|\\
\leq&\frac{1}{\beta}\mathbb{E}_{x\sim\rho,y\sim\pi^{\mathrm{E}}(\cdot|x)}\left\|\sum_{\tilde{y}\in\mathcal{A}}\left[\nabla_{\boldsymbol{\theta}}\log\frac{\pi_{\boldsymbol{\theta}_2}(y|x)}{\pi_{\mathrm{ref}}(y|x)}-\nabla_{\boldsymbol{\theta}}\log\frac{\pi_{\boldsymbol{\theta}_2}(\tilde{y}|x)}{\pi_{\mathrm{ref}}(\tilde{y}|x)}\right](\pi_{\boldsymbol{\theta}_1}(\tilde{y}|x)-\pi_{\boldsymbol{\theta}_2}(\tilde{y}|x))\right\|\\
\leq&\frac{1}{\beta}\frac{2L_0}{\pi_{\min}}\mathbb{E}_{x\sim\rho,y\sim\pi^{\mathrm{E}}(\cdot|x)}\left\|\sum_{\tilde{y}\in\mathcal{A}}(\pi_{\boldsymbol{\theta}_1}(\tilde{y}|x)-\pi_{\boldsymbol{\theta}_2}(\tilde{y}|x))\right\|\\
=&\frac{1}{\beta}\frac{2L_0}{\pi_{\min}}\mathbb{E}_{x\sim\rho,y\sim\pi^{\mathrm{E}}(\cdot|x)}\left\|\sum_{\tilde{y}\in\mathcal{A}}\pi_{\boldsymbol{\theta}_1}(\tilde{y}|x)\frac{\pi_{\boldsymbol{\theta}_1}(\tilde{y}|x)-\pi_{\boldsymbol{\theta}_2}(\tilde{y}|x)}{\pi_{\boldsymbol{\theta}_1}(\tilde{y}|x)}\right\|\leq\frac{1}{\beta}\frac{2L_0^2}{\pi_{\min}^2}\|\boldsymbol{\theta}_1-\boldsymbol{\theta}_2\|
\end{aligned}
$$

Plugging these back to (16) we get

$$
\|\nabla\ell(\boldsymbol{\theta}_1)-\nabla\ell(\boldsymbol{\theta}_2)\|\leq\frac{2}{\beta}\left(\frac{\pi_{\max}L_1+2L_0^2}{\pi_{\min}^2}\right)\|\boldsymbol{\theta}_1-\boldsymbol{\theta}_2\|
$$

$\square$

Now since we generate at the beginning of the inner loop, the estimator $\hat{\nabla}\ell(\boldsymbol{\theta}_{t,k})$ is not an unbiased estimator of $\nabla\ell(\boldsymbol{\theta}_{t,k})$ for any $k>0$, i.e.

$$
\nabla\ell(\boldsymbol{\theta}_{t,k})=\frac{1}{\beta}\mathbb{E}_{(x_{t,k},y_{t,k})\sim\mathcal{D},\tilde{y}_{t,k}\sim\pi_{\boldsymbol{\theta}_{t,k}}(\cdot|x_{t,k})}\left[\nabla_{\boldsymbol{\theta}}\log\frac{\pi_{\boldsymbol{\theta}_{t,k}}(y_{t,k}|x_{t,k})}{\pi_{\mathrm{ref}}(y_{t,k}|x_{t,k})}-\nabla_{\boldsymbol{\theta}}\log\frac{\pi_{\boldsymbol{\theta}_{t,k}}(\tilde{y}_{t,k}|x_{t,k})}{\pi_{\mathrm{ref}}(\tilde{y}_{t,k}|x_{t,k})}\right]
$$
(17)

$$
\neq\mathbb{E}\hat{\nabla}\ell(\boldsymbol{\theta}_{t,k})=\frac{1}{\beta}\mathbb{E}_{(x_{t,k},y_{t,k})\sim\mathcal{D},\tilde{y}_{t,k}\sim\pi_{\boldsymbol{\theta}_{t,0}}(\cdot|x_{t,k})}\left[\nabla_{\boldsymbol{\theta}}\log\frac{\pi_{\boldsymbol{\theta}_{t,k}}(y_{t,k}|x_{t,k})}{\pi_{\mathrm{ref}}(y_{t,k}|x_{t,k})}-\nabla_{\boldsymbol{\theta}}\log\frac{\pi_{\boldsymbol{\theta}_{t,k}}(\tilde{y}_{t,k}|x_{t,k})}{\pi_{\mathrm{ref}}(\tilde{y}_{t,k}|x_{t,k})}\right]
$$
(18)

We thus need to carefully analyze this biasedness so that the convergence can be boosted by a large $K$, since a large $K$ will result in a very large bias.

Now we are ready to re-state and prove Theorem 3.1:

**Theorem B.1.** *Suppose Assumption B.1 holds, then for Algorithm 1 and 2 with $\eta_t=\Theta(1/\sqrt{TK})$ we have*

$$
\min_{t=1,\ldots,T,\,k=1,\ldots,K}\mathbb{E}[\|\nabla\ell(\boldsymbol{\theta}_{t,k})\|^2]\leq\mathcal{O}\left(\frac{\Delta_0+LG^2}{\sqrt{TK}}+\frac{\tilde{L}^2G^2}{T}\right)
$$

*where $\Delta_0=\ell^*-\ell(\boldsymbol{\theta}_0)$ and we omit constant factors in $\tilde{\mathcal{O}}$.*

**Proof.** We prove directly for Algorithm 2 since the gradient estimator (12) and the estimator $g_{t,k}$ Algorithm 1 (we do solve the $\pi$ subproblem to its optimum) are both for the original bilevel problem (4).

From the Lipschitz gradient of $\ell$ we have

$$
\ell(\boldsymbol{\theta}_{t,k+1})\geq\ell(\boldsymbol{\theta}_{t,k})+\eta_t\langle\hat{\nabla}\ell(\boldsymbol{\theta}_{t,k}),\nabla\ell(\boldsymbol{\theta}_{t,k})\rangle-\frac{\eta_t^2L}{2}\|\hat{\nabla}\ell(\boldsymbol{\theta}_{t,k})\|^2
$$

i.e.

$$
\eta_t\|\nabla\ell(\boldsymbol{\theta}_{t,k})\|^2\leq(\ell(\boldsymbol{\theta}_{t,k+1})-\ell(\boldsymbol{\theta}_{t,k}))+\eta_t\langle\nabla\ell(\boldsymbol{\theta}_{t,k})-\hat{\nabla}\ell(\boldsymbol{\theta}_{t,k}),\nabla\ell(\boldsymbol{\theta}_{t,k})\rangle+\frac{\eta_t^2L}{2}\|\hat{\nabla}\ell(\boldsymbol{\theta}_{t,k})\|^2
$$

Taking expectation to $\boldsymbol{\theta}_{t,k}$ and by Assumption B.2, we have

$$
\eta_t\mathbb{E}\|\nabla\ell(\boldsymbol{\theta}_{t,k})\|^2\leq(\mathbb{E}\ell(\boldsymbol{\theta}_{t,k+1})-\ell(\boldsymbol{\theta}_{t,k}))+\eta_t\langle\nabla\ell(\boldsymbol{\theta}_{t,k})-\mathbb{E}\hat{\nabla}\ell(\boldsymbol{\theta}_{t,k}),\nabla\ell(\boldsymbol{\theta}_{t,k})\rangle+\frac{\eta_t^2LG^2}{2}
$$

where the expectation is taken w.r.t. the sample $\tilde{y}_{t,k}$ to generate the estimator of current iteration.

Sum up from $k = 0$ to $k = K$ we get

$$\sum_{k=0}^{K-1} \eta_t \mathbb{E}\|\nabla\ell(\boldsymbol{\theta}_{t,k})\|^2 \le (\mathbb{E}\ell(\boldsymbol{\theta}_{t,K-1}) - \ell(\boldsymbol{\theta}_{t,0})) + \eta_t \sum_{k=0}^{K-1} \langle \nabla\ell(\boldsymbol{\theta}_{t,k}) - \mathbb{E}\hat{\nabla}\ell(\boldsymbol{\theta}_{t,k}), \nabla\ell(\boldsymbol{\theta}_{t,k})\rangle + \frac{\eta_t^2 LG^2 K}{2} \tag{19}$$

Since the expectation is taken only on the random sample at current iteration, and we know that the true gradient and the approximated gradient are (17) and (18), we have the following estimate:

$$\|\nabla\ell(\boldsymbol{\theta}_{t,k}) - \mathbb{E}\hat{\nabla}\ell(\boldsymbol{\theta}_{t,k})\|$$

$$= \frac{1}{\beta}\left\| \mathbb{E}_{x_{t,k}\sim\rho, y_{t,k}\sim\pi^{\mathrm{E}}(\cdot|x_{t,k}), \tilde{y}_{t,k}\sim\pi_{\boldsymbol{\theta}_{t,k}}(\cdot|x_{t,k})}\left[ \nabla_{\boldsymbol{\theta}}\log\frac{\pi_{\boldsymbol{\theta}_{t,k}}(y_{t,k}|x_{t,k})}{\pi_{\mathrm{ref}}(y_{t,k}|x_{t,k})} - \nabla_{\boldsymbol{\theta}}\log\frac{\pi_{\boldsymbol{\theta}_{t,k}}(\tilde{y}_{t,k}|x_{t,k})}{\pi_{\mathrm{ref}}(\tilde{y}_{t,k}|x_{t,k})} \right] \right.$$

$$\left. - \mathbb{E}_{x_{t,k}\sim\rho, y_{t,k}\sim\pi^{\mathrm{E}}(\cdot|x_{t,k}), \tilde{y}_{t,k}\sim\pi_{\boldsymbol{\theta}_{t,k}}(\cdot|x_{t,k})}\left[ \nabla_{\boldsymbol{\theta}}\log\frac{\pi_{\boldsymbol{\theta}_{t,k}}(y_{t,k}|x_{t,k})}{\pi_{\mathrm{ref}}(y_{t,k}|x_{t,k})} - \nabla_{\boldsymbol{\theta}}\log\frac{\pi_{\boldsymbol{\theta}_{t,k}}(\tilde{y}_{t,k}|x_{t,k})}{\pi_{\mathrm{ref}}(\tilde{y}_{t,k}|x_{t,k})} \right] \right\|$$

$$= \frac{1}{\beta}\left\| \mathbb{E}_{x_{t,k}, y_{t,k}} \int \left[ \nabla_{\boldsymbol{\theta}}\log\frac{\pi_{\boldsymbol{\theta}_{t,k}}(y_{t,k}|x_{t,k})}{\pi_{\mathrm{ref}}(y_{t,k}|x_{t,k})} - \nabla_{\boldsymbol{\theta}}\log\frac{\pi_{\boldsymbol{\theta}_{t,k}}(\tilde{y}|x_{t,k})}{\pi_{\mathrm{ref}}(\tilde{y}|x_{t,k})} \right] \left( \pi_{\boldsymbol{\theta}_{t,k}}(\tilde{y}|x_{t,k}) - \pi_{\boldsymbol{\theta}_{t,0}}(\tilde{y}|x_{t,k}) \right) d\tilde{y} \right\|$$

$$\le \frac{1}{\beta}\frac{2L_0^2}{\pi_{\min}}\|\boldsymbol{\theta}_{t,k} - \boldsymbol{\theta}_{t,0}\|$$

Denote $\tilde{L} = \frac{1}{\beta}\frac{2L_0^2}{\pi_{\min}}$, we thus have:

$$\sum_{k=0}^{K-1} \langle \nabla\ell(\boldsymbol{\theta}_{t,k}) - \mathbb{E}\hat{\nabla}\ell(\boldsymbol{\theta}_{t,k}), \nabla\ell(\boldsymbol{\theta}_{t,k})\rangle \le \sum_{k=0}^{K-1} \|\nabla\ell(\boldsymbol{\theta}_{t,k}) - \mathbb{E}\hat{\nabla}\ell(\boldsymbol{\theta}_{t,k})\|\|\nabla\ell(\boldsymbol{\theta}_{t,k})\|$$

$$\le \frac{1}{2}\sum_{k=0}^{K-1} \|\nabla\ell(\boldsymbol{\theta}_{t,k}) - \mathbb{E}\hat{\nabla}\ell(\boldsymbol{\theta}_{t,k})\|^2 + \frac{1}{2}\sum_{k=0}^{K-1} \|\nabla\ell(\boldsymbol{\theta}_{t,k})\|^2$$

$$\le \frac{\tilde{L}^2}{2}\sum_{k=0}^{K-1}\|\boldsymbol{\theta}_{t,k} - \boldsymbol{\theta}_{t,0}\|^2 + \frac{1}{2}\sum_{k=0}^{K-1}\|\nabla\ell(\boldsymbol{\theta}_{t,k})\|^2 = \frac{\eta_t^2\tilde{L}^2}{2}\sum_{k=0}^{K-1}\left\|\sum_{i=0}^{k-1}\hat{\nabla}\ell(\boldsymbol{\theta}_{t,i})\right\|^2 + \frac{1}{2}\sum_{k=0}^{K-1}\|\nabla\ell(\boldsymbol{\theta}_{t,k})\|^2$$

Therefore

$$\sum_{k=0}^{K-1} \langle \nabla\ell(\boldsymbol{\theta}_{t,k}) - \mathbb{E}\hat{\nabla}\ell(\boldsymbol{\theta}_{t,k}), \nabla\ell(\boldsymbol{\theta}_{t,k})\rangle \le \frac{\eta_t^2\tilde{L}^2 G^2}{2}\frac{K(K-1)}{2} + \frac{1}{2}\sum_{k=0}^{K-1}\|\nabla\ell(\boldsymbol{\theta}_{t,k})\|^2$$

Substituting back into (19) leads to

$$\frac{1}{2}\sum_{k=0}^{K-1}\eta_t\mathbb{E}\|\nabla\ell(\boldsymbol{\theta}_{t,k})\|^2 \le (\mathbb{E}\ell(\boldsymbol{\theta}_{t,K-1}) - \ell(\boldsymbol{\theta}_{t,0})) + \eta_t^3\tilde{L}^2 G^2\frac{K(K-1)}{2} + \frac{\eta_t^2 LG^2 K}{2}$$

Summing up from $t = 0$ to $T - 1$ gives

$$\frac{1}{2}\sum_{t=0}^{T-1}\sum_{k=0}^{K-1}\eta_t\mathbb{E}\|\nabla\ell(\boldsymbol{\theta}_{t,k})\|^2 \le \mathbb{E}\ell(\boldsymbol{\theta}_{T-1,K-1}) - \ell(\boldsymbol{\theta}_{-1,0}) + \sum_{t=0}^{T-1}\eta_t^3\tilde{L}^2 G^2\frac{K(K-1)}{2} + \sum_{t=0}^{T-1}\frac{\eta_t^2 LG^2 K}{2}$$

With a constant step size $\eta_t = \eta > 0$, we have

$$\frac{1}{2TK}\sum_{t=0}^{T-1}\sum_{k=0}^{K-1}\mathbb{E}\|\nabla\ell(\boldsymbol{\theta}_{t,k})\|^2 \le \frac{\mathbb{E}\ell(\boldsymbol{\theta}_{T-1,K-1}) - \ell(\boldsymbol{\theta}_{-1,0})}{\eta TK} + \eta^2\tilde{L}^2 G^2\frac{K-1}{2} + \eta\frac{LG^2}{2}$$

Taking $\eta = \Theta(1/\sqrt{TK})$, we get

$$\frac{1}{TK}\sum_{t=0}^{T-1}\sum_{k=0}^{K-1}\mathbb{E}\|\nabla\ell(\boldsymbol{\theta}_{t,k})\|^2 = \mathcal{O}\left( \frac{\mathbb{E}\ell(\boldsymbol{\theta}_{T-1,K-1}) - \ell(\boldsymbol{\theta}_{-1,0}) + LG^2}{\sqrt{TK}} + \frac{\tilde{L}^2 G^2}{T} \right)$$

Hence as $T \to \infty$, the rate is $\mathcal{O}(1/\sqrt{TK})$. $\qquad\square$

## C   Implementation details of the numerical experiments

We follow the code as in SPIN [Chen et al., 2024], where we utilize DeepSpeed ZeRO-3 [Rajbhandari et al., 2020] and FlashAttention-2 [Dao, 2023] to reduce the memory cost. We use RMSProp [Hinton et al., 2012] optimizer with no weight decay. For 1b models, we use two NVIDIA A100-40G to do the training with per device batch size of 4 for Algorithm 1 and per device batch size of 8 for Algorithm 2. For 7b models we use eight NVIDIA A100-40G to do the training with per device batch size of 2. We train all models with bfloat16 prevision. We set the peak learning rate to be 5e-7 for first two epochs and 1e-7 for the next two epochs. We fix $\beta = 0.1$ and consider the max sequence length to be 1024 for 1b models and 2048 for 7b models. We use the same prompt template "### Instruction: prompt\n\n### Response: " as in Chen et al. [2024]. For the policy optimization step in Algorithm 1, we use the PPO trainer in the TRL package [von Werra et al., 2020]. For the HuggingFace Open LLM Leaderboard evaluation, we use the Language Model Evaluation Harness library (v0.4.2) [Gao et al., 2023], and we also use the same number of few-shots as in Chen et al. [2024].

Finally, in Table 5, we further provide the generation examples of our fine-tuned model in Table 4.

## D   Additional numerical results based on LoRA

During the reviewing and discussion periods, we conducted extra experiments with LoRA to provide further evidences for this work.

We first provide the result of 7b experiments with LoRA in Table 6. In this setting we see a significant improvement over the pretrained model (zephyr-7b-sft-full), where we observe 2.3% lift from the baseline and a 1% lift from SPIN. SFT in contrast can only achieve less than 1% lift from the base line. The reason here might be due to the limited model size when using LoRA, a contrastive training better helps the model distinguishing the referred and non-preferred continuations, yielding better performance over the standard SFT.

As a side note, we do not anticipate to significantly outperform SPIN since algorithmically our proposed IRFT method includes SPIN as a special case. Rather, one of our main objective is to provide a theoretical foundation for the contrastive-type training algorithm, such as SPIN, which can all be studied under the bilevel inverse RL framework. The comparison with SPIN largely indicates that SPIN is still a RL-based fine-tuning method, suggesting an alternative interpretation that leads to provable convergence in lieu of the two-player game interpretation in Chen et al. [2024].

**Reconciling our result with SPIN [Chen et al., 2024]** Readers may realize that the result in Section 5.3 is different from SPIN [Chen et al., 2024]. We believe that this is due to two reasons:

1. First, the baseline in our paper is different from SPIN paper: We believe a different baseline model is used in SPIN as evidenced in the Github discussions and the SPIN paper was released before the baseline is fully trained (Jan 2 vs Jan 10)[2]. In particular, in Table 3 of the SPIN paper, the base model yields a "26.76" accuracy for GSM8k dataset, but we observe "31.92" which is significantly higher. We notice that a newer version of both the model zephyr-7b-sft-full[3] and the lm_eval evaluation package[4] which both our paper and SPIN use for evaluation are used in our paper. We run test on different versions of base model and eval codebase and obtain Table 7, where we indeed see that the new version of the base model has a significant lift in the performance on GSM8k comparing to the old version. We remark that SPIN paper observes the most significant increase of SPIN algorithm on GSM8k task (from 26.76 to 35.10). **Since we use the newest model in all our experiments, it leaves much less space for us to improve from.**

2. Second, we should not compare the iter3's 8.63% increase in Table 3 of SPIN with our paper's 2.66% increase directly. When we say we take $T = 5$ and epoch=2 as in Table 4, we essential split the data into 5 chunks and generate more frequently than SPIN, but still consume and generate for all the training data for 2 epochs in total (SPIN iter0 also trained for 2 epochs). So what we need to compare is the first iteration of SPIN (which is SPIN

---

[2]see for example the discussion in this link.
[3]See their model commit history.
[4]See their codebase.

Table 5: Generation example of fine-tuned models in Table 4.

| Prompt | Craft an intriguing opening paragraph for a fictional short story. The story should involve a character who wakes up one morning to find that they can time travel. |
|---|---|
| `zephyr-7b-sft-full` | It was just another ordinary morning for Emily, until she opened her eyes and found herself in a different time and place. She was lying on her bed, surrounded by strange decor and people dressed in clothes from a bygone era. Emily rubbed her eyes, thinking she must be dreaming. But as she got up, she realized that she could feel the weight of the day ahead of her, much like she always did. It was only then that she realized that she had the power to time travel. |
| IRFT $T = 1$ (SPIN iter 0) | As soon as her eyes fluttered open, something was different. The room looked the same, the sun streaming in through the same window, but something felt off. It wasn't until she tried to sit up that she realized what it was - her body felt lighter, as if she were floating on air. Confused, she looked around, searching for an explanation, and that's when she saw it - the clock on her bedside table was ticking backwards. Panic set in as she tried to sit up, but her body refused to obey her commands. And just like that, she was hurled through time, landing with a thud in a completely different era. |
| IRFT $T = 1$ (SPIN iter 1) | As soon as the alarm clock shrilled, something felt different to James, but he brushed it aside and reached out to turn it off, just like any other morning. Yet, as soon as he sat up, he felt a strange pull as if he was being dragged backwards, and his room began to spin in a dizzying whirlwind of colors. In the blink of an eye, he found himself standing in a different place altogether, and he knew that he had just woken up in a different time. |
| IRFT $T = 5$ (2 epochs) | I opened my eyes groggily, ready to start another mundane day. But as I rubbed my eyes and stretched my arms, something strange happened. I felt a sudden jolt of energy course through my veins, and before I knew it, the world around me had transformed. The room that I was in was no longer my own, and I was standing in the middle of a bustling medieval town square. I blinked in disbelief, trying to make sense of what had just happened. Had I gone mad, or was I dreaming? But as I looked around me, I realized with awe that I had somehow managed to time travel. |
| IRFT $T = 5$ (4 epochs) | It was just another regular morning for Emily, until as soon as she opened her eyes the room started to blur and spin. Panicking, she tried to stand up and grab the nightstand but her hand went right through it. Confused and terrified, she tried to scream, but as soon as the sound started to come out of her throat, she was engulfed by a bright light that covered her body. When Emily opened her eyes again, she realized that she had traveled back in time, to a moment when she was 9 years old and standing at the foot of her parents' bed, ready to tell them the good news about acing her history test. |

iter0 in SPIN's original paper). Table 4 essentially indicates that, under our fair comparison setting, every iteration of our proposed algorithm outperforms every iteration of SPIN.

Last, we provide a simple experiment to show how accurate the reward learned by our model is. This experiment also addresses the generalization ability of the reward. Since we train our 7b model with a high-quality dataset (ultrachat) and we believe that the corresponding implicit reward $r = \log(\pi_{\boldsymbol{\theta}}/\pi_{\text{ref}})$ should already be pretty accurate in terms of distinguish the good over the rejected continuations. Therefore we did a simple test: we construct the implicit reward by

Table 6: Test performance of SPIN [Chen et al., 2024] and IRFT (Algorithm 2) based on `zephyr-7b-sft-full` across HuggingFace Open LLM Leaderboard datasets. In this table we test with LoRA ($r = 64$).

| Tasks
Metrics | T | K | AI2_Arc
acc_norm | TruthfulQA
acc | Winogrande
acc | GSM8k
exact_match | HellaSwag
acc_norm | MMLU
acc | Average |
|---|---|---|---|---|---|---|---|---|---|
| `zephyr-7b-sft-full` | 0 | 0 | 74.83 | 34.07 | 76.09 | 31.92 | 81.09 | 58.86 | 59.48 |
| SFT | NA | NA | 75.20 | 34.18 | 76.16 | 34.95 | 80.96 | 57.71 | 59.86 |
| IRFT (SPIN) | 1 | $\frac{\text{\# samples}}{\text{batchsize}} * 2$ | 75.31 | 35.67 | 75.85 | 34.5 | 81.98 | 57.46 | 60.13 |
| IRFT | 5 | $\frac{\text{\# samples}}{\text{batchsize}} * \frac{2}{5}$ | 74.92 | 37.96 | 76.95 | 35.25 | 82.48 | 57.66 | **60.87** |

Table 7: Test performance of different versions of the base model `zephyr-7b-sft-full`.

| Version | lm_eval version | Arc_challenge | TruthfulQA_mc2 | Winogrande | GSM8k | HellaSwag | MMLU | Average |
|---|---|---|---|---|---|---|---|---|
| Newest | v0.4.0 | 58.02 | 40.40 | 76.16 | 34.19 | 80.89 | 57.46 | 57.85 |
| Version in SPIN | v0.4.0 | 60.84 | 43.74 | 78.69 | 26.23 | 82.79 | 58.97 | 58.54 |

equation $r = \log(\pi_{\boldsymbol{\theta}}/\pi_{\text{ref}})$ where we compare different $\pi_{\boldsymbol{\theta}}$ (pretrained, SFT, SPIN and IRFT) on the ultrafeedback dataset (note that we did not do training on this dataset) which is a preference dataset. We believe that the accurate reward model $r = \log(\pi_{\boldsymbol{\theta}}/\pi_{\text{ref}})$ should be able to distinguish the preferred and rejected continuations, and we compute the ratio of $r(\text{preferred}) > r(\text{rejected})$ (which is called win-rate in some literature) and obtain Table 8, where we can see that IRFT improves the implicit reward's distinguishability of chosen over rejected.

Table 8: Win-rate of models trained by different methods. In this table we test with LoRA ($r = 64$).

| Model | SFT | SPIN (IRFT $T = 1$) | IRFT $T = 5$ |
|---|---|---|---|
| Win-rate | 42.6% | 42.8% | 55.6% |

