# OpenReview forum: "Getting More Juice Out of the SFT Data: Reward Learning from Human Demonstration Improves SFT for LLM Alignment"
_NeurIPS.cc/2024/Conference — NeurIPS 2024 poster_

### Official Review · Reviewer_Tyd8 · 2024-07-09

**Soundness:** 3
**Presentation:** 3
**Contribution:** 2
**Rating:** 7
**Confidence:** 4

**Summary:**

This paper proposes an approach to aligning Large Language Models (LLMs) with human preferences by integrating Inverse Reinforcement Learning (IRL) into the supervised fine-tuning (SFT) stage. The proposed IRL-based method simultaneously builds a reward model and a policy model during the SFT stage, enhancing efficiency and robustness against low-quality data.

**Strengths:**

1. The paper introduces a novel IRL-based method for the SFT stage, providing a fresh perspective on improving LLM alignment.

2. The paper provides a strong theoretical basis for the proposed algorithms, ensuring that they converge to stationary solutions of the IRL problem.

3. The evaluation includes both theoretical analysis and empirical testing on large-scale models, ensuring the validity of the results.

**Weaknesses:**

1. The paper does not address the scalability of the proposed methods to even larger models or different types of LLMs beyond the ones tested.

2. The evaluation results only present the empirical performance but lack experiments on computation costs introduced by this method.

**Questions:**

1. It seems that RFT and IRFT perform equally. What's the takeaway from these two methods? Moreover, how to choose them wisely.
2. Typo in L226, there's an error in referring to the Appendix.
3. In L204-206 "In 204 practice however we take a relatively small T and large K, because frequent on-line sampling is time 205 consuming" How does this parameter choice affect the final performance, especially compared to large T and Large K?

**Limitations:**

The limitations are included in the conclusion section.

---

> ### Author Rebuttal · Authors · 2024-08-05
>
> > The paper does not address the scalability of the proposed methods to even larger models or different types of LLMs beyond the ones tested.
>
> **Response**: We thank the reviewer for the suggestion. Here we believe provide a theoretical analysis would be enough since we know exactly how much memory and time each of the proposed algorithm would need comparing to SFT. For Algorithm 1 in the paper, we need to maintain a reward model and a policy model (which is the LLM), and this is doubling the standard LLM fine-tuning. In short, the memory consumption and computation time of Algorithm 1 is similar to the standard RLHF process, where we also have the reward learning and policy optimization processes; For Algorithm 2, we simply need to maintain the policy (LLM) model, and the memory consumption would be exactly the **same** as the standard SFT, whereas the computation time would involving generating for the entire training sample, which would be of similar level as the standard RLHF process.
> We summarize the memory consumption and the computational time of the proposed methods in the table below, *assuming that the reward and policy models are of same size*. Here *Forward* means the memory required for storing a model in inference mode, and *Backward* is the memory required for storing a model in training mode, including weights, activations and gradients; also *SFT* means the computational time as standard SFT, and *Generation* means the time to generate continuations for each of the input training prompts. *2SFT+Generation* is roughly the same time as standard RLHF.
>
> | Method | Peak Memory    | Computational Time |
> | ------- | -------- | ------- |
> | Algorithm 1 | Forward+Backward  | 2SFT+Generation |
> | Algorithm 2 | Backward | SFT+Generation |
>
> > The evaluation results only present the empirical performance but lack experiments on computation costs introduced by this method.
>
> **Response**: We thank the reviewer for the question. We refer to the answer to the previous question, where we do include the computational time analysis of the proposed methods. In short, Algorithm 1 is similar to RLHF, and Algortihm 2 is similar to a generation plus a DPO process.
>
> > It seems that RFT and IRFT perform equally. What's the takeaway from these two methods? Moreover, how to choose them wisely.
>
> **Response**: The difference between RFT and IRFT are similar to that of RLHF and DPO. RFT produces an explicit reward model for further generalization and IRFT is more memory and time efficient. We advocate using IRFT if you are not looking for an explicit reward model from the demonstration dataset (see our general response for the detail of the generalization ability of the reward model).
>
> > Typo in L226, there's an error in referring to the Appendix.
>
> **Response**: We thank the reviewer for the reminder. We modfied it correspondingly.
>
> > In L204-206 "In 204 practice however we take a relatively small T and large K, because frequent on-line sampling is time 205 consuming" How does this parameter choice affect the final performance, especially compared to large T and Large K?
>
> **Response**: We did our test on a large T and small K, i.e. we generate samples for every training batch and do the update. The performance is largely similar to the SFT performance (less than 1% lift from the baseline), as shown below (see our general rebuttal to all reviewers for the implementation details):
> | Task     | Arc | TruthfulQA | Winogrande | GSM8k | Hellaswag | MMLU | Average |
> | -------- | ------- | ------- | ------- | ------- | ------- | ------- | ------- |
> | zephyr-7b-sft-full  | 74.83    | 34.07    | 76.09    | 31.92    | 81.09    | 58.86    | 59.48    |
> | SFT | 75.20    | 34.18    | 76.16    | 34.95    | 80.96    | 57.71    | 59.86    |
> | IRFT (T=1, SPIN)    | 75.31    | 35.67    | 75.85    | 34.5    | 81.98    | 57.46    | 60.13    |
> | IRFT (T=5)    | 74.92    | 37.96    | 76.95    | 35.25    | 82.48    | 57.66    | **60.87**    |
> | IRFT (T=#batch)    | 75.23    | 33.58    | 75.37    | 33.13    | 82.26    | 57.68    | 59.54    |
>
> where T=#batch is the case when $T$ is large and $K$ is small.
>
> In short, we believe that generating too frequently would not only be time-consuming, but also detrimental to the model performance (since we take to much variance into account at each stochastic gradient step for generation). We would recommend a reasonable frequency in generation which leads to the optimal performance, such as what we observed when we generate 5 to 10 times for the entire training dataset.

---

> > ### Comment · Reviewer_Tyd8 · 2024-08-12
> >
> > Thank you for your clear response, I will keep my positive score.

---

### Official Review · Reviewer_RhUZ · 2024-07-11

**Soundness:** 3
**Presentation:** 3
**Contribution:** 3
**Rating:** 6
**Confidence:** 4

**Summary:**

This paper proposes to study if IRL techniques can be used for aligning large language models. They propose two different algorithms: one that explicitly learns a reward model and one that implicitly learns the reward in the policy. These reward models are learned by contrasting expert generations and the policy generations.

**Strengths:**

- The paper writing and motivation is clear.
- The paper includes in depth theoretical analysis.
- They evaluate the trained LLMs on a wide variety of benchmarks.

**Weaknesses:**

- The experiments are not very convincing. For the Open LLM leaderboard experiments, the gain is really small (around 1 or 2%). It seems likely that this gain is due to variance in the training process, especially since IRFT has ~1% variance in different hyperparameter settings.
- There isn’t much discussion of the computational cost of the algorithms.

**Questions:**

- How accurate is the learned reward model?
- The experimental setting is not very clear. Is Algorithm 1 used on the top 10K data selected by the reward model? If so this is really problematic: shouldn’t you fine tune on the whole dataset (or a randomly downsampled subset)? Can you assume that you will have access to 10K samples with high reward scores? And wouldn't just be better to use the reward model used for data selection for RL?
- How robust is IRFT to hyperparameter setting compared to SFT?
- Is 1-2% increase on the OpenLLM benchmark meaningful?

**Limitations:**

Yes, the authors discuss limitations.

---

> ### Author Rebuttal · Authors · 2024-08-05
>
> > The experiments are not very convincing. For the Open LLM leaderboard experiments, the gain is really small (around 1 or 2%). It seems likely that this gain is due to variance in the training process, especially since IRFT has ~1% variance in different hyperparameter settings.
>
> **Response**: To further show the strength of our algorithms, we tested two extra experiments and inclide the results in the general rebuttal to all reviewers. First, we test our algorithm for 7b models with parameter-efficient fine-tuning (PEFT) settings and show that the proposed method could yield a **2.3% improvement** comparing to the baselines. Additionally, we also conduct a reward accuracy analysis in the general rebuttal where we show that the implicit reward learned by our method is more accurate than that of SFT and SPIN (please also see our answer to your third question). We hope this could give more strong evidence on the superiority of the proposed method.
>
> > There isn’t much discussion of the computational cost of the algorithms.
>
> **Response**: We thank the reviewer for bringing this issue up. We refer to the general rebuttal to all reviewers where we include a detailed computational cost analysis.
>
> > How accurate is the learned reward model?
>
> **Response**: We conduct a very simple experiment to show how accurate the reward learned by our model is. Since we train our 7b model with a high-quality dataset (**ultrachat**) and the corresponding implicit reward $r=\log(\frac{\pi_{\mathbf{\theta}}}{\pi_{\text{ref}}})$ should already be pretty accurate in terms of distinguish the good over the rejected continuations. Therefore we construct the implicit reward by equation $r=\log(\frac{\pi_{\mathbf{\theta}}}{\pi_{\text{ref}}})$ where we compare different $\pi_{\mathbf{\theta}}$ (pretrained, SFT, SPIN and IRFT) on the **ultrafeedback** dataset (note that we **did not do training on this dataset**) which is a preference dataset. We believe that the accurate reward model $r=\log(\frac{\pi_{\mathbf{\theta}}}{\pi_{\text{ref}}})$ should be able to distinguish the preferred and rejected continuations, and we compute the ratio of $r(\text{preferred})>r(\text{rejected})$ and obtain the follow table:
> | Model     | SFT | SPIN (IRFT with $T=1$) | IRFT ($T=5$) |
> | -------- | ------- | ------- | ------- |
> | Ratio $r(\text{preferred})>r(\text{rejected})$ | 42.6% | 42.8% | 55.6% |
>
> where we can see that IRFT **significantly improves the implicit reward's distinguishability of chosen over rejected**. We hope this addresses the reviewer's concern.
>
> > The experimental setting is not very clear. Is Algorithm 1 used on the top 10K data selected by the reward model? If so this is really problematic: shouldn’t you fine tune on the whole dataset (or a randomly downsampled subset)? Can you assume that you will have access to 10K samples with high reward scores? And wouldn't just be better to use the reward model used for data selection for RL?
>
> **Response**: We thank the reviewer for the question. For the experiment on Algorithm 1, we want to demonstrate that with **high quality** demonstration data, our proposed Algorithm 1 is more capable of learning an accurate reward model that is able to distinguish the good over the rejected continuations, by assigning higher scores to the good continuations. Note that we need to construct a "high quality dataset" at the first place. The experiment setting is that we use a well-established reward model (beaver-7b-v3.0-reward) to pick up good demonstration data (which are the top 10k data in terms of reward score), and **fairly** train the model with SFT and our method on these 10k data, then compute the rewards of the continuations generated by SFT trained model and our trained model.
>
> We are not assuming we always have access to top 10k samples, instead, top 10k sample is used as a relatively simple way to obtain high quality data to verify our presumption that our method is more capable of getting better reward if we have good demonstration data.
>
> In practice, the implication of this experiment is that, if you have good demonstration data (under certain score criteria) at hand and would like your language model to align with it, then our proposed method should be more capable of extract this underlying score criteria and better aligning with it. **We do not assume that we always have a well-established reward model on hand in practice**.
>
> > How robust is IRFT to hyperparameter setting compared to SFT?
>
> **Response**: We thank the reviewer for the question and we believe IRFT is robust to hyperparameters in general. We test all our experiment of IRFT, SPIN and SFT with the same learning rate, epochs and batchsize. We do have new hyperparameters $T$ and $K$. However when you fix $T$, total epochs and the training dataset, $K$ is automatically determined. Therefore the only parameter one needs to determine in extra is $T$. We found that in practice taking $T=5$ (or multiples of 5, meaning training for more epochs) would result in the best performance for both 1b and 7b models. In short, the only extra parameter we have is $T$ and we believe $T=5$ is readily a good candidate for most of the alignment tasks.
>
> > Is 1-2% increase on the OpenLLM benchmark meaningful?
>
> **Response**: Yes, we do believe that it is meaningful. The openLLM leaderboard benchmark is one of the most heavily-used (SPIN also primarily used this) influential benchmark in the community where it does not rely on large models such as GPT4 but only on multiple downstream tasks to test the performance of different LLMs. For models like Llama3-70B, they in general also only achieve only 1 to 2% increase over the previous state-of-the-art (if you check their hugginface webpage, they also tested on tasks such as MMLU, GSM8K and Winogrande). Therefore we believe we are making meaningful progress using the same tasks as SPIN on 7b models.

---

> > ### Comment · Reviewer_RhUZ · 2024-08-08
> > **Concern about OpenLLM benchmark and Comparison to SPIN**
> >
> > Thank you for providing more details on the computational cost of the algorithm and showing the accuracy of the obtained reward model. I am still concerned over the weak performance on the OpenLLM leaderboard. I checked the SPIN paper, and they reported a 8-9% improvement on the OpenLLM leaderboard. Do you know why there is a large discrepancy between the results reported in SPIN and those in this paper? Furthermore, is it possible to re-run some of your experiments with different seeds and conduct significance tests on the results?

---

> > > ### Author Response · Authors · 2024-08-10
> > >
> > > > **Concern about OpenLLM benchmark and Comparison to SPIN**:
> > >
> > > Thanks for checking the rebuttal. TL;DR: 1. we suspect that a different version of baseline "zephyr-7b-sft-full" has been used in the SPIN paper which led to a significantly higher lift of 8-9% improvement, compared to ours of 2.6%. 2. when evaluated on the same baseline and codebase, every iteration our proposed algorithm outperforms every iteration of SPIN.
> > >
> > > The SPIN paper reported the following improvements:
> > >
> > > *(base model: older version; evaluation codebase: v0.4.0; test without LoRA)*
> > > | Iterations  |  zephyr-7b-sft-full   | iter0 | iter1 | iter2 | iter3 |
> > > | -------- | ------- | ------- | ------- | ------- | ------- |
> > > | Average Performance | 58.14 | 60.80 | 62.12 | 62.97 | 63.16 |
> > > | Absolute increase |  | 2.66 | 1.32 | 0.85 | 0.89 |
> > > | Relative increase |  | 4.56% | 6.85% | 8.31% | 8.63% |
> > >
> > > We believe the result should be interpreted in the following way:
> > > 1. First, the baseline in our paper is different from SPIN paper: We believe a different baseline model is used in SPIN as evidenced in the Github discussions & the SPIN paper was released before the baseline is fully trained (Jan 2 vs Jan 10), see for example [this discussion](https://github.com/uclaml/SPIN/issues/12). In particular, in Table 3 of the SPIN paper, the base model yields a "26.76" accuracy for GSM8k dataset, but we observe "31.92" which is significantly higher. We notice that a newer version of both the model zephyr-7b-sft-full (see [their model commit history](https://huggingface.co/alignment-handbook/zephyr-7b-sft-full/commits/main)) and the lm_eval evaluation package which both our paper and SPIN use for evaluation (see [their codebase](https://github.com/EleutherAI/lm-evaluation-harness)) are used in our paper. We run test on different versions of base model and eval codebase and obtain a table as follows:
> > > *(Performance of different version of zephyr-7b-sft-full, note we also change the first two tasks to make it fully aligned with SPIN paper)*
> > > | Base model version| Eval package version     | Arc_challenge | TruthfulQA_mc2 | Winogrande | GSM8k | Hellaswag | MMLU | Average |
> > > | -------- | -------| ------- | ------- | ------- | ------- | ------- | ------- | ------- |
> > > | [Newest](https://huggingface.co/alignment-handbook/zephyr-7b-sft-full)  | [v0.4.0](https://github.com/EleutherAI/lm-evaluation-harness/tree/v0.4.0) |  58.02  |  40.40   |  76.16   |  34.19   |  80.89   |   57.46  | 57.85   |
> > > | [Version in SPIN](https://huggingface.co/alignment-handbook/zephyr-7b-sft-full/commit/90e0792328bc522e1662a3a7c611b030d563bf5b)  | [v0.4.0](https://github.com/EleutherAI/lm-evaluation-harness/tree/v0.4.0) |  60.84 | 43.74  |  78.69  |  26.23   |  82.79   |  58.97   | 58.54   |
> > >
> > > where we indeed see that the new version of the base model has a significant lift in the performance on GSM8k comparing to the old version. We remark that SPIN paper observes the most significant increase of SPIN algorithm on GSM8k task (from 26.76 to 35.10). **Since we use the newest model in all our experiments, it leaves much less space for us to improve from.**
> > >
> > > 2. Second, we should not compare the iter3's 8.63% increase with our paper's 2.66% increase directly. When we say we take $T=5$ and epoch=2 as in Table 3 of our paper, we essential split the data into 5 chunks and generate more frequently than SPIN, but still consume and generate for all the training data for **2 epochs in total** (SPIN iter0 also trained for 2 epochs). So what we need to compare is the first iteration of SPIN (which is SPIN iter0 in SPIN's original paper). In view of Table 3 in our paper, a fair comparison is the following table, note that the baseline is newest version of the model on the newest evaluation codebase (we follow the "iteration" notion in SPIN paper):
> > >
> > > *(base model: newest version; evaluation codebase: v0.4.3; test without LoRA)*
> > > | Iterations  |  zephyr-7b-sft-full   | iter0 | iter1 |
> > > | -------- | ------- | ------- | ------- |
> > > | SPIN | 59.48 | 60.32 | 61.02 |
> > > | IRFT | 59.48 | 60.71 | 61.03 |
> > >
> > > Essentially **under our fair comparison setting**, every iteration of our proposed algorithm outperforms every iteration of SPIN. We didn't further do experiment for iter2 and iter3 because under our experiment setting, both SPIN and IRFT seem to have less significant improvement from iter0 to iter1 (less than 1 point in average accuracy), and also due to our limited computing resources.
> > >
> > > (To be continued)

---

> > > > ### Author Response · Authors · 2024-08-10
> > > >
> > > > (Continuing above)
> > > >
> > > > We also tried to revert back to v0.4.0 of the evaluation codebase (which we believe should be the version of evaluation codebase used by SPIN) and strictly following the metrics SPIN paper mentioned. We have the following result for the same setting as the Table in our general rebuttal:
> > > >
> > > > *(base model: newest version; evaluation codebase: v0.4.0; test with LoRA r=64)*
> > > > | Task     | Arc_challenge | TruthfulQA_mc2 | Winogrande | GSM8k | Hellaswag | MMLU | Average |
> > > > | -------- | ------- | ------- | ------- | ------- | ------- | ------- | ------- |
> > > > | zephyr-7b-sft-full  |  58.02  |  40.4   |  76.16   |  34.19   |  80.89   |   57.46  | 57.85   |
> > > > | SFT |  58.87   |  40.34   |  76.56   | 34.5   |  81.42   | 57.93    |   58.27  |
> > > > | IRFT (T=1, SPIN iter0)    |  59.64   |   40.4  |  75.85   | 34.5    |  81.98   |  57.46   |  58.31   |
> > > > | IRFT (T=5, iter0)    | 61.09    |  42.92   | 76.95   |  35.25   |  82.48   |  57.66   |  59.39   |
> > > >
> > > > In the above table, the baseline performancec in GSM8k is "34.19", which is still significantly higher than "26.76" in that of SPIN paper. Therefore the baseline is still significantly different from SPIN paper, also our algorithm (last row) is still outperforming SPIN (second last row) by more than 1 point of average accuracy in iter0. We hope this clarification makes the comparison clear. We will point this out and revise our experiment setting section accordingly.
> > > >
> > > > > Furthermore, is it possible to re-run some of your experiments with different seeds and conduct significance tests on the results?
> > > >
> > > > **Response**: Due to limited computational resources, we did not conduct extensive experiment on multiple seeds (note that SPIN also did not do significance test). Here we provide 3 trials on IRFT iter0 (with T=5) with three different seeds:
> > > >
> > > > *(base model: newest version; evaluation codebase: v0.4.3; test without LoRA)*
> > > > | Task     | Arc | TruthfulQA | Winogrande | GSM8k | Hellaswag | MMLU | Average |
> > > > | -------- | ------- | ------- | ------- | ------- | ------- | ------- | ------- |
> > > > | zephyr-7b-sft-full  | 74.83 | 34.07 | 76.09 | 31.92 | 81.09 | 58.86 | 59.48 |
> > > > | IRFT (T=5, iter0) trial1   |  75.82 | 39.99 | 77.19 | 31.24 | 82.07 | 57.93 | 60.71
> > > > | IRFT (T=5, iter0) trial2   |  76.66 |	36.25 |	76.48 |	32.37 |	82.13 |	58.30|	60.37  |
> > > > | IRFT (T=5, iter0) trial3   |  75.42 |	35.85 |	76.80 |	32.60 |	82.27 |	58.92 |	60.31  |
> > > >
> > > > Above table indicates a variance of 0.05 (a standard deviation of 0.22) for the average performance. We wish this gives you a flavor of the level of variance for our experiments.
> > > >
> > > > It might be very hard for us to systematically test the random seed effect starting from now. For each 7b experiment, we need to generate continuation for each prompt, which will take roughly 8 hours for the entire training dataset and another 8 hours to do the full-fine-tune on 8 A100-40G; For LoRA (r=64) we need 4 A100-40G for a similar time. We would need to wait in the queue for resources, which makes it impossible to estimate the time for us to finish. The same situation applies for running the SPIN algorithm.

---

> > > > > ### Comment · Reviewer_RhUZ · 2024-08-10
> > > > > **Thank you for the discussion**
> > > > >
> > > > > Thank you for the comprehensive experiments and discussion of the SPIN codebase. All of my concerns are cleared up and I will raise my score.

---

> > > > > > ### Author Response · Authors · 2024-08-11
> > > > > >
> > > > > > Thank you very much for re-evaluating our work and we will incorporate the discussion to the revised paper!

---

### Official Review · Reviewer_J5fv · 2024-07-13

**Soundness:** 2
**Presentation:** 2
**Contribution:** 3
**Rating:** 5
**Confidence:** 3

**Summary:**

The authors propose using inverse reinforcement learning (IRL) on demonstration data in place of supervised learning, as is typically done for LLMs. The intuition is that human preferences are also encoded in demonstrations collected for SFT. Concretely, the authors propose a bilevel optimization approach with policy learning at the lower level and reward learning at the upper level. Two methods are proposed for alignment: with either implicit or explicit reward learning. The authors demonstrate improved performance when used to finetune smaller language models, compared to existing SFT approaches.

**Strengths:**

- The method seems (to the best of my knowledge) to be novel and interesting, tying together recent LLM literature and IRL methods.
- The authors compare against relevant baselines (normal SFT, or SPIN) across a suite of benchmarks and two model sizes.
- The overarching idea and question the paper strives to answer is of importance and relevance to the research community, as the proposed methods enable extracting more value out of SFT data.

**Weaknesses:**

- The experimental results seem to support the efficacy of the proposed algorithms, however the gains across various tasks with IRFT seem fairly marginal (it would help if Table 2 and 3 contained error bars, perhaps with policies finetuned with different seeds). As it stands, it seems like different benchmarks results are sensitive to different hyperparameters (choice of T), and the baselines outperform IRFT on 3/6 of the benchmarks in Table 3, suggesting that the additional complexity of the method does not necessarily lead to improved performance across all tasks. For example, the best average performance of IRFT is 61.03, whereas the best baseline achieves 61.02.
- Furthermore, the results seem more mixed in Table 3 compared to Table 2, suggesting that it's not clear the method is still effective as model sizes increase.
- It would help to present the SFT results in Table 3. Even though the authors note that further SFT could degrade performance, it would help to see how effectively using the demonstration data with IRFT or SPIN compares against naive SFT.
- Minor note: line 157 seems incomplete, "even when the demonstration policy is"...(extreme)?
- Minor note: the notation is a bit confusing, with theta referring to both the model parameters and the reward model parameters. For clarity, it would help if separate variables were used.
- Minor note: missing reference to appendix in line 226.
- Minor note: grammar error in line 320 -- "SPIN and IRFT are both capable of further improv(ing) the performance.."

**Questions:**

See Weaknesses section for suggestions.

**Limitations:**

The paper is missing more thorough discussion on the method’s limitations and weaknesses (e.g. sensitivity to T, more complicated training process compared to SFT, etc.).

---

> ### Author Rebuttal · Authors · 2024-08-05
>
> > The experimental results seem to support the efficacy of the proposed algorithms, however the gains across various tasks with IRFT seem fairly marginal (it would help if Table 2 and 3 contained error bars, perhaps with policies finetuned with different seeds). As it stands, it seems like different benchmarks results are sensitive to different hyperparameters (choice of T), and the baselines outperform IRFT on 3/6 of the benchmarks in Table 3, suggesting that the additional complexity of the method does not necessarily lead to improved performance across all tasks. For example, the best average performance of IRFT is 61.03, whereas the best baseline achieves 61.02.
>
> **Response**: To further show the strength of our algorithms, we tested our algorithm for 7b models with parameter-efficient fine-tuning (PEFT) settings and show that the proposed method could yield a **2.3% improvement** comparing to the baselines. The result tables are in our general rebuttal to all reviewers. We hope this could give more strong evidence on the superiority of the proposed method.
>
> > Furthermore, the results seem more mixed in Table 3 compared to Table 2, suggesting that it's not clear the method is still effective as model sizes increase.
>
> **Response**: Again please see our new results included in the general rebuttal to all reviewers. First, we show that for the LoRA setting, our proposed method significantly outperforms SFT. Additionally, we also conduct a reward accuracy analysis in the general rebuttal where we show that the implicit reward learned by our method is more accurate than that of SFT and SPIN. We believe the extra evidence should further support that our proposed method yields significant improvements over SFT.
>
> > It would help to present the SFT results in Table 3. Even though the authors note that further SFT could degrade performance, it would help to see how effectively using the demonstration data with IRFT or SPIN compares against naive SFT.
>
> **Response**: We thank the reviewer for this suggestion. Our claim "SFT could degrade performance" is largely from [1] page 8 "SFT on further epochs 2 and 3 fails to yield more than 1% improvement". We also verified this during the rebuttal period. Due to the limit time during the rebuttal period, we were only able to conduct the experiment on LoRA and the result can be see in the general rebuttal to all reviewers, where we indeed see **less than 1% lift of SFT from the baseline**, verifying the claim made by [1].
> We will rephrase "SFT could degrade performance" to "SFT on further epochs 2 and 3 fails to yield more than 1% improvement" in the revised paper.
>
> > Minor note: line 157 seems incomplete, "even when the demonstration policy is"...(extreme)?
>
> **Response**: We thank the reviewer for the reminder. We meant to say "even when the demonstration policy is extreme" and will modify it accordingly.
>
> > Minor note: the notation is a bit confusing, with theta referring to both the model parameters and the reward model parameters. For clarity, it would help if separate variables were used.
>
> **Response**: We thank the reviewer for the reminder. For RFT (Algorithm 1), the $\theta$ is the parameter for reward since the policy is determined by tge reward; For IRFT (Algorithm 2) $\theta$ is the parameter for the policy. We will modify the $\theta$ in Algorithm 1 into $\phi$ in our revised paper.
>
> > Minor note: missing reference to appendix in line 226.
>
> **Response**: We thank the reviewer for the reminder. We modfied it correspondingly.
>
> > Minor note: grammar error in line 320 -- "SPIN and IRFT are both capable of further improv(ing) the performance.."
>
> **Response**: We thank the reviewer for the reminder. We modfied it correspondingly.
>
> > The paper is missing more thorough discussion on the method’s limitations and weaknesses (e.g. sensitivity to T, more complicated training process compared to SFT, etc.).
>
> **Response**: We thank the reviewer for raising this issue up. We believe IRFT is robust to hyperparameters in general. We test all our experiment of IRFT, SPIN and SFT with the same learning rate, epochs and batchsize. We do have new hyperparameters $T$ and $K$. However when you fix $T$, total epochs and the training dataset, $K$ is automatically determined. Therefore the only parameter one needs to determine in extra is $T$. We found that in practice taking $T=5$ (or multiples of 5, meaning training for more epochs) would result in the best performance for both 1b and 7b models. In short, the only extra parameter we have is $T$ and we believe $T=5$ is readily a good candidate for most of the alignment tasks.
>
> Please refer to our general rebuttal to all authors for the computational cost analysis.
>
> **References**:
>
> [1] Chen, Zixiang, et al. "Self-play fine-tuning converts weak language models to strong language models." International Conference on Machine Learning (2024).

---

> > ### Comment · Reviewer_J5fv · 2024-08-12
> >
> > I thank the authors for responding to my questions. The additional experiments on the accuracy of the implicit reward model are interesting to see, and overall the proposed method presents an interesting perspective on introducing IRL in the SFT stage. However, the improvements in the LoRA setting do not fully address my concerns around the lack of clear improvement coming from the method. As such, I have adjusted my score accordingly.

---

### Official Review · Reviewer_AMst · 2024-07-30

**Soundness:** 2
**Presentation:** 3
**Contribution:** 2
**Rating:** 5
**Confidence:** 3

**Summary:**

This paper proposes two methods focusing on RLHF, namely RFT and IRFT. The takeaway message is that the SFT stage also significantly benefits from learning a reward model instead of using the human demonstration data directly via supervised learning.

**Strengths:**

1. The paper is clear, illustrating the differences between similar works.
2.  The motivation is clear, and the method is reasonable. Most importantly, it provides parts of theoretical analysis for the convergence.

**Weaknesses:**

1. The training procedure is complicated, meaning there are many issues in tunning. Is there any computation cost analysis for better understanding the limitations of this method?
2. Though the author discusses SPIN, the truth is that equation 4.7 in SPIN is very similar to equation 6 in this paper.
3. The experimental results are not strong. It did not surpass the SPIN by a large margin. Are there any other baselines that can be incorporated such as DPO?

**Questions:**

How can the reward model obtain the generalization power? Can the author say more about it, in my understanding, the main issue that affects the generalization is the scale of the demonstration dataset.

**Limitations:**

See weaknesses. I have to say I am not an expert in this field, therefore, I am willing to change my attitude if I find I am wrong.

---

> ### Author Rebuttal · Authors · 2024-08-05
>
> > 1. The training procedure is complicated, meaning there are many issues in tunning. Is there any computation cost analysis for better understanding the limitations of this method?
>
> **Response**: We thank the reviewer for bringing this issue up. For Algorithm 1 in the paper, we need to maintain a reward model and a policy model (which is the LLM), and this is doubling the standard LLM fine-tuning. In short, the memory consumption and computation time of Algorithm 1 is similar to the standard RLHF process, where we also have the reward learning and policy optimization processes; For Algorithm 2, we simply need to maintain the policy (LLM) model, and the memory consumption would be exactly the **same** as the standard SFT, whereas the computation time would involving generating for the entire training sample, which would be of similar level as the standard RLHF process.
> We summarize the memory consumption and the computational time of the proposed methods in the table below, *assuming that the reward and policy models are of same size*. Here *Forward* means the memory required for storing a model in inference mode, and *Backward* is the memory required for storing a model in training mode, including weights, activations and gradients; also *SFT* means the computational time as standard SFT, and *Generation* means the time to generate continuations for each of the input training prompts. *2SFT+Generation* is roughly the same time as standard RLHF.
>
> | Method | Peak Memory    | Computational Time |
> | ------- | -------- | ------- |
> | Algorithm 1 | Forward+Backward  | 2SFT+Generation |
> | Algorithm 2 | Backward | SFT+Generation |
>
> > 2. Though the author discusses SPIN, the truth is that equation 4.7 in SPIN is very similar to equation 6 in this paper.
>
> **Response**: As we stated from Line 251 page 7, our Algorithm 2 include the algorithm in SPIN as a special case and our inverse reinforcement learning framework naturally introduces a difference of log probabilities formulation. We believe that the claim "the truth is that ... is very similar" is not very accurate, but instead we would say that the algorithms **coincide under certain conditions**. There is still a key difference between (4.7) in SPIN and our equation 6: in our equation 6 we take the negative sample $\tilde{y}$ generated by our current model $p_{\mathbf{\theta}}$, whereas in SPIN the negative sample $\tilde{y}$ is sampled from the model from the based model of current training epoch $p_{\mathbf{\theta}_t}$.
>
> > 3. The experimental results are not strong. It did not surpass the SPIN by a large margin. Are there any other baselines that can be incorporated such as DPO?
>
> **Response**: We don't think it necessary to compare with DPO. As we made it clear in the abstract, our proposed framwork is a method for the supervised fine-tune stage of alignment, which means that we do not have access to a preference dataset but only a demonstration dataset. Direct preference optimization (DPO) is not designed for aligning demonstration dataset. In fact, our proposed method can be regarded as a DPO-type algorithm for the supervised fine-tune stage. So we cannot compare our method fairly with DPO, at least not when DPO utilizes preference data.
> However, in order to further show the strength of our algorithms, we tested our algorithm for 7b models with additional parameter-efficient fine-tuning (PEFT) settings and show that the proposed method could yield a **2.3% improvement** comparing to the baselines. The result tables are in our general rebuttal to all reviewers. We hope this could give more strong evidence on the superiority of the proposed methods.
>
> > 4. How can the reward model obtain the generalization power? Can the author say more about it, in my understanding, the main issue that affects the generalization is the scale of the demonstration dataset.
>
> **Response**: The power of this proposed method is that we can learn a reward model even without the preference dataset, and we completely agree with the reviewer on this point that the generalization power relies on the scale and quality of the demonstration dataset. For example, we train our 7b model with a high-quality dataset (**ultrachat**) and we believe that the corresponding implicit reward $r=\log(\frac{\pi_{\mathbf{\theta}}}{\pi_{\text{ref}}})$ should already obtain certain generalization ability. Therefore we did a simple test below to show the generalization power of the reward model learned from the proposed algorithm: we construct the implicit reward by equation $r=\log(\frac{\pi_{\mathbf{\theta}}}{\pi_{\text{ref}}})$ where we compare different $\pi_{\mathbf{\theta}}$ (SFT, SPIN and IRFT) on the **ultrafeedback** dataset (note that we **did not do training on this dataset**) which is a preference dataset. We believe that the reward model $r=\log(\frac{\pi_{\mathbf{\theta}}}{\pi_{\text{ref}}})$ with a better generalization ability should be able to distinguish the preferred and rejected continuations, and we compute the ratio of $r(\text{preferred})>r(\text{rejected})$ and obtain the follow table:
> | Model     | SFT | SPIN (IRFT with $T=1$) | IRFT ($T=5$) |
> | -------- | ------- | ------- | ------- |
> | Ratio $r(\text{preferred})>r(\text{rejected})$ | 42.6% | 42.8% | 55.6% |
>
> where we can see that IRFT **significantly inproves the implicit reward's distinguishability of chosen over rejected, indicating a superior generalization ability.**

---

> > ### Comment · Reviewer_AMst · 2024-08-12
> >
> > Thanks to the author's rebuttal, most of my doubts have been resolved, but I still don't think the performance gap is significant. In the discussion phase, I would discuss this point. No matter what the discussion result is, my final score will align with other reviewers.

---

### Author Rebuttal · Authors · 2024-08-05

We thank all the reviewers for valuable comments. In this general rebuttal, we provide answers and results to questions that are raised by multiple reviewers. We wish our extra analysis and results could help you better understanding the contribution of the work, which we believe is significant to the community.

# Additional experiments to show the effectiveness of proposed algorithms

## LoRA experiments
We first sincerely apologize for an important typo in our initial draft. In Table 3, we claimed that we use LoRA for 7b models. This is not true since **we actually did a full fine-tune** with the proposed method. We did use LoRA to validate the method but did not collect the results. We now collect the correct results where we test our algorithm with LoRA (r=64) as follows:
| Task     | Arc | TruthfulQA | Winogrande | GSM8k | Hellaswag | MMLU | Average |
| -------- | ------- | ------- | ------- | ------- | ------- | ------- | ------- |
| zephyr-7b-sft-full  | 74.83    | 34.07    | 76.09    | 31.92    | 81.09    | 58.86    | 59.48    |
| SFT | 75.20    | 34.18    | 76.16    | 34.95    | 80.96    | 57.71    | 59.86    |
| IRFT (T=1, SPIN)    | 75.31    | 35.67    | 75.85    | 34.5    | 81.98    | 57.46    | 60.13    |
| IRFT (T=5)    | 74.92    | 37.96    | 76.95    | 35.25    | 82.48    | 57.66    | **60.87**    |

We compared the standard SFT, SPIN (IRFT with T=1) and IRFT with T=5. In this setting we see a significant improvement over the pretrained model (zephyr-7b-sft-full), where we observe **2.3% lift from the baseline and a 1% lift from SPIN**. SFT in contrast can only achieve less than 1% lift from the base line. The reason here might be due to the limited model size when using LoRA, a constrastive training better helps the model distinguishing the referred and non-preferred continuations, yielding better performance over the standard SFT.

As a side note, we **do not** anticipate to significantly outperform SPIN since algorithmically our proposed IRFT method includes SPIN as a **special case**. Rather, one of our main objective is to provide a theoretical foundation for the constrastive-type training algorithm, such as SPIN, which can all be studied under the bilevel inverse RL framework. The comparison with SPIN largely indicates that SPIN is still a RL-based fine-tuning method, suggesting an alternative interpretation that leads to provable convergence in lieu of the two-player game interpretation in [1].

## The accuracy/generalization ability of the learned reward model
We conduct a simple experiment to show how accurate the reward learned by our model is. This experiment also addresses the generalization ability of the reward. Since we train our 7b model with a high-quality dataset (**ultrachat**) and we believe that the corresponding implicit reward $r=\log(\frac{\pi_{\mathbf{\theta}}}{\pi_{\text{ref}}})$ should already be pretty accurate in terms of distinguish the good over the rejected continuations. Therefore we did a simple test: we construct the implicit reward by equation $r=\log(\frac{\pi_{\mathbf{\theta}}}{\pi_{\text{ref}}})$ where we compare different $\pi_{\mathbf{\theta}}$ (pretrained, SFT, SPIN and IRFT) on the **ultrafeedback** dataset (note that we **did not do training on this dataset**) which is a preference dataset. We believe that the accurate reward model $r=\log(\frac{\pi_{\mathbf{\theta}}}{\pi_{\text{ref}}})$ should be able to distinguish the preferred and rejected continuations, and we compute the ratio of $r(\text{preferred})>r(\text{rejected})$ and obtain the follow table:
| Model     | SFT | SPIN (IRFT with $T=1$) | IRFT ($T=5$) |
| -------- | ------- | ------- | ------- |
| Ratio $r(\text{preferred})>r(\text{rejected})$ | 42.6% | 42.8% | 55.6% |

where we can see that IRFT **significantly improves the implicit reward's distinguishability of chosen over rejected**. We hope this addresses the reviewer's concern.

# Computational cost analysis
For Algorithm 1 in the paper, we need to maintain a reward model and a policy model (which is the LLM), and this is doubling the standard LLM fine-tuning. In short, the memory consumption and computation time of Algorithm 1 is similar to the standard RLHF process (RLHF=reward learning + policy optimization); For Algorithm 2, we simply need to maintain the policy (LLM) model, and the memory consumption would be exactly the **same** as the standard SFT, whereas the computation time would involving generating for the entire training sample, which would be of similar level as the standard policy optimization process (same computational time as SPIN). Note that standard policy optimization process is equivalent to the time of standard SFT and a generation process toward all training input prompts.

We thus summarize the memory consumption and the computational time of the proposed methods in the table below, *assuming that the reward and policy models are of same size*. Here *Forward* means the memory required for storing a model in inference mode, and *Backward* is the memory required for storing a model in training mode, including weights, activations and gradients; also *SFT* means the computational time as standard SFT, and *Generation* means the time to generate continuations for each of the input training prompts. *2SFT+Generation* is roughly the same time as standard RLHF.

| Method | Peak Memory    | Computational Time |
| ------- | -------- | ------- |
| Algorithm 1 | Forward+Backward  | 2SFT+Generation |
| Algorithm 2 | Backward | SFT+Generation |

Reference:

[1] Chen, Zixiang, et al. "Self-play fine-tuning converts weak language models to strong language models." International Conference on Machine Learning (2024).

---

### Author Response · Authors · 2024-08-11
**A gentle reminder on the discussion period**

Dear reviewers,

We thank all of your constructive feedbacks toward this work, and we tried our best to address your concerns and questions. As the author-reviewer discussion period is ending in three days, we gently remind you to check out our rebuttal and let us know if you still have any comments. Thank you very much!

Best,
Authors of submission 14483

---

### Decision · Program_Chairs · 2024-09-25

**Decision:**

Accept (poster)

**Comment:**

This work proposes a novel large language model (LLM) alignment approach using Reward Learning from Human Demonstration. All reviewers consistently recommended accepting this work. AC agrees that this work is interesting and deserves to be published on NeurIPS 2024. The reviewers did raise some valuable concerns that should be addressed in the final camera-ready version of the paper. The authors are encouraged to make the necessary changes in the final version.